# CPL: Critical Plan Step Learning Boosts LLM Generalization in Reasoning Tasks

## Abstract

Post-training, particularly reinforcement learning (RL) using self-play-generated data, has become a new learning paradigm for large language models (LLMs). However, scaling RL to develop a general reasoner remains a research challenge, as existing methods focus on task-specific reasoning without adequately addressing generalization across a broader range of tasks. Moreover, unlike traditional RL with limited action space, LLMs operate in an infinite space, making it crucial to search for valuable and diverse strategies to solve problems effectively. To address this, we propose searching within the action space on high-level abstract plans to enhance model generalization and introduce Critical Plan Step Learning (CPL), comprising: 1) searching on plan, using Monte Carlo Tree Search (MCTS) to explore diverse plan steps in multi-step reasoning tasks, and 2) learning critical plan steps through Step-level Advantage Preference Optimization (Step-APO), which integrates advantage estimates for step preference, obtained via MCTS, into Direct Preference Optimization (DPO). This combination helps the model effectively learn critical plan steps, enhancing both reasoning capabilities and generalization. Experimental results demonstrate that our method, trained exclusively on GSM8K and MATH, not only significantly improves performance on GSM8K (+10.5%) and MATH (+6.5%), but also enhances out-of-domain reasoning benchmarks, such as HumanEval (+12.2%), GPQA (+8.6%), ARC-C (+4.0%), MMLU-STEM (+2.2%), and BBH (+1.8%). The code is available at https://anonymous.4open.science/r/CPL.

## 1 Introduction

Large language models (LLMs) have achieved significant success through scaling, particularly in pre-training on vast datasets (OpenAI et al., 2024; Dubey et al., 2024). Recently, there has been an increasing focus on scaling post-training, especially through reinforcement learning (RL) on self-play-generated data, which has emerged as a new learning paradigm for LLMs. Notably, OpenAI's o1 (OpenAI, 2024) has consistently improved its reasoning abilities through large-scale RL, which teaches the model to think more productively. Additionally, recent research works (Xie et al., 2024; Feng et al., 2023; Chen et al., 2024) leverage Monte Carlo Tree Search (MCTS) (Kocsis & Szepesvári, 2006) to iteratively collect preference data. RL on this self-generated data facilitates iterative self-improvement, leading to significantly enhanced reasoning capabilities.

However, scaling RL to develop a general reasoner remains a research challenge. Traditional RL methods, such as AlphaGo (Silver et al., 2016), struggle with generalization due to their specific action spaces. Existing approaches for LLMs primarily focus on enhancing task-specific or domain-specific reasoning capabilities, such as in mathematics or coding. While this has led to significant improvements in these specific tasks, it has not adequately addressed the model's generalization abilities across various reasoning tasks. Furthermore, unlike traditional RL, which operates in a limited action space, LLMs function within a vast search space. This expansive scope, combined with the high inference latency of LLMs, limits both the diversity and quality of explored reasoning paths.

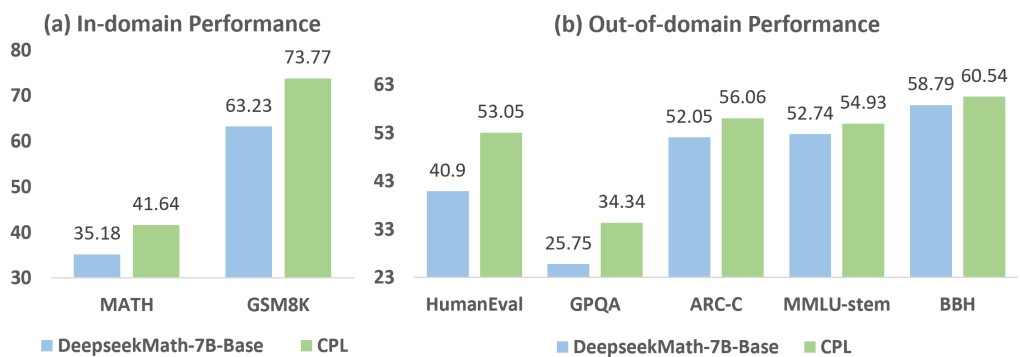

Figure 1: Overview of CPL results. (a) In-Domain Performance: Our CPL-trained model significantly outperforms the DeepSeekMath-7B-Base on in-domain tasks. (b) Out-of-Domain Performance: Our CPL method also shows strong generalization, outperforming the baseline model on out-of-domain reasoning tasks.

To enhance generalization in reasoning tasks, we propose searching within the action space on high-level abstract plans, rather than focusing on task-specific action spaces that often limit generalization. Building on previous work (Wang et al., 2023; Yao et al., 2023; Hao et al., 2023) that uses LLMs to generate both reasoning plans and task-specific solutions to boost reasoning capabilities, we argue that task-specific solutions—like mathematical formulas, code or symbolic solutions—are closely tied to task-specific skills. In contrast, plans represent abstract thinking for problem-solving, such as determining which knowledge to apply or how to break down a problem, helping models develop broader, task-agnostic abilities that improve generalization (illustrated in Figure 2 Left).

Furthermore, under the challenge of a vast search space of reasoning paths, we propose that maintaining diversity and identifying critical paths are essential for solving complex problems. Plan-based search enables better exploration of high-level strategies and can achieve better diversity; whereas, solutions-based search may limit diversity, as different solutions may share the same underlying thought. Besides, existing preference-learning methods, such as Direct Preference Optimization (DPO) (Rafailov et al., 2023) faces challenges in learning critical steps due to their inability to capture fine-grained supervision. Recent works propose Step-level DPO (Setlur et al., 2024; Lai et al., 2024) to learn step-level preferences, but its reliance on heuristics, such as marking the first error step as dispreferred, limits full exploration of the search space and model optimization. To address this, we propose a method to identify and learn critical plan steps that provide higher advantages for improving the model's reasoning ability (illustrated in Figure 2 Right).

Thus, we introduce Critical Plan Step Learning (CPL), which consists of two key components:

1. Searching on plan, specifically, we devise a step-by-step plan to solve the problem, with the final step providing the full solution based on the plan. Using MCTS to explores diverse plan steps in multi-step reasoning tasks, it creates a plan tree, where high-quality plan step preferences are derived from the final outcome. This process enables the exploration of high-level strategies, helping the model acquire task-agnostic skills and improve generalization across different tasks.

2. Learning critical plan steps through Step-level Advantage Preference Optimization (Step-APO), which builds upon DPO. Step-APO integrates advantage estimates for step-level preference pairs obtained via MCTS, enabling the model to learn fine-grained preferences between steps, identify critical plan steps, and de-emphasize erroneous ones.

To conclude, our contributions are: 1) We explore the scaling problem in RL and propose searching within the action space on high-level abstract plans to enhance model generalization, rather than focusing on task-specific action spaces that often limit generalization. 2) We introduce a novel approach CPL, which leverages MCTS to explore diverse plan steps, distinguishing it from existing methods that focus on exploring solutions, and uses our Step-APO to learn step-level plan preferences, thereby helping the model effectively identify and

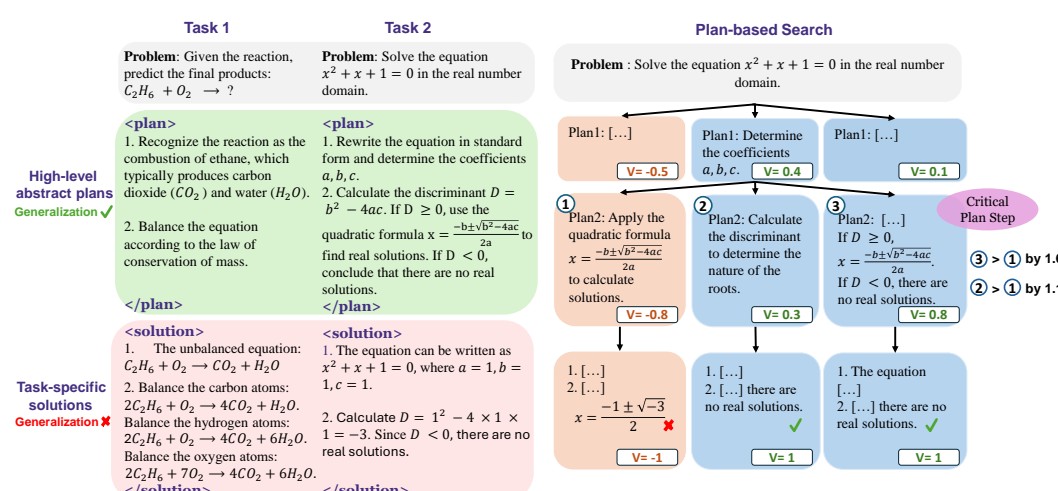

Figure 2: Illustration of CPL. Left: Plans represent abstract thinking for problem-solving, which allows for better generalization, whereas task-specific solutions often limit it. Right: CPL searches within the action space on high-level abstract plans using MCTS and obtains advantage estimates for step-level preferences. CPL can then identify and learn critical steps that provide a clear advantage over others.

learn critical steps. 3) Extensive experiments show that CPL enhances reasoning capabilities and generalization across tasks, achieving significant improvements in both in-domain and out-of-domain tasks, as shown in Figure 1.

## 2 METHODS

In this section, we introduce our Critical Plan Step Learning (CPL), it boosts model performance via iterative process over plan-based search and step-level preference learning. We first introduce our plan-based MCTS, which enables the LLM to explore diverse plan strategies in the vast search space. Next, we present our Step-APO in detail to further explore the potential of step-level preference learning in multi-step reasoning task. Finally, we describe how we iteratively optimize the policy model and value model.

### 2.1 PLAN-BASED MCTS

MCTS builds a reasoning tree iteratively and autonomously explores step-level reasoning traces, which can be used to optimize LLMs. Existing methods (Chen et al., 2024; Xie et al., 2024) that leverage MCTS to collect data for training usually focus on exploring solution steps within the entire search space or on simultaneously exploring both plans and solutions. To improve transfer performance across a broader range of reasoning tasks, we propose learning high-level abstract plans, which enables the model to acquire more task-agnostic capabilities and thereby achieve better generalization. We first create a step-by-step plan to solve the problem, with the final step presenting the full solution and final answer based on the plan. The prompt is provided in the Appendix A. Ultimately, we obtain a plan tree and high-quality plan step supervision through iterative search with MCTS.

Specifically, given the plan tree $\mathcal{T}$, each node represents a state $\mathbf{s}_t$, and each edge represents an action $\mathbf{a}_t$, which corresponds to a reasoning step that leads to the next state $\mathbf{s}_{t+1}$. Under the same parent node, different sibling nodes form a set of step-level preference pairs, with each node having its own value $V(\mathbf{s}_t)$ representing the expected future reward under state $\mathbf{s}_t$. These values can be obtained through the MCTS process, which involves four key operations: selection, expansion, evaluation, and backup. To enhance efficiency, we use a value model to assess the expected returns from the partial reasoning paths, with the final integration of

both policy and value models guiding the search process. Next, we describe the four steps of MCTS.

**Selection**: We use the PUCT algorithm (Rosin, 2011) to guide the selection process with the following formula, where $N$ represents the visit count:

$$\arg\max_{\mathbf{a}_t} \left[ Q(\mathbf{s}_t, \mathbf{a}_t) + c_{\text{puct}} \pi_\theta(\mathbf{a}_t | \mathbf{s}_t) \frac{\sqrt{N(\mathbf{s}_t)}}{1 + N(\mathbf{s}_t, \mathbf{a}_t)} \right]. \tag{1}$$

**Expansion and Evaluation**: During expansion, we sample multiple possible candidate actions for the next step. During evaluation, the terminal node is assessed by comparing the final answer with the ground truth, while the values of other nodes are predicted by the value model.

**Backup**: Once a terminal node is reached, we perform a bottom-up update from the terminal node back to the root. We update the visit count $N$, the state value $V$, and the transition value $Q$ as follows:

$$Q(\mathbf{s}_t, \mathbf{a}_t) \leftarrow r(\mathbf{s}_t, \mathbf{a}_t) + V(\mathbf{s}_{t+1}), \tag{2}$$

$$V(\mathbf{s}_t) \leftarrow \sum_a N(\mathbf{s}_{t+1}) Q(\mathbf{s}_t, \mathbf{a}_t) / \sum_a N(\mathbf{s}_{t+1}), \tag{3}$$

$$N(\mathbf{s}_t) \leftarrow N(\mathbf{s}_t) + 1. \tag{4}$$

## 2.2 STEP-APO TO LEARN CRITICAL PLAN STEPS

Unlike mainstream approaches (Hwang et al., 2024; Lai et al., 2024) that learn step-level preferences by identifying the first error step and sampling a corresponding preferred step, while potentially yielding more accurate preferences, this method lacks sufficient exploration of the vast reasoning trace space. Given the large variations in advantage differences across different data pairs, we propose Step-APO, which introduces advantage estimates for preference pairs into DPO. This enables the model to more effectively learn critical intermediate plan steps, thereby further improving its reasoning capabilities. Next, We will provide its derivation and analysis from the perspective of its gradient.

### 2.2.1 PRELIMINARIES

**The Classical RL Objective** RLHF approaches (Ziegler et al., 2020; Bai et al., 2022; Ouyang et al., 2022) usually first learn a reward function from human feedback, then optimize it with a policy gradient-based method like PPO (Schulman et al., 2017) with an entropy-bonus using the following multi-step RL objective:

$$\max_{\pi_\theta} \mathbb{E}_{\mathbf{a}_t \sim \pi_\theta(\cdot | \mathbf{s}_t)} \left[ \sum_{t=0}^{T} (r(\mathbf{s}_t, \mathbf{a}_t) + \underbrace{\beta \log \pi_{\text{ref}}(\mathbf{a}_t | \mathbf{s}_t)}_{\text{KL penalty}}) + \beta \mathcal{H}(\pi_\theta) | \mathbf{s}_0 \sim \rho(\mathbf{s}_0) \right], \tag{5}$$

where $r(\mathbf{s}_t, \mathbf{a}_t)$ denotes the step-level reward function, followed by a KL penalty that aims to ensure the learned policy $\pi_\theta$ does not deviate significantly from the reference policy $\pi_{\text{ref}}$. $\pi_{\text{ref}}$ is typically produced via supervised fine-tuning.

**Direct Preference Optimization** DPO (Rafailov et al., 2023) uses the well-known closed-form optimal solution, which establishes a mapping between the reward model and the optimal policy under the KL divergence, obtaining the reward as:

$$r(\mathbf{x}, \mathbf{y}) = \beta \log \pi^*(\mathbf{y} | \mathbf{x}) - \beta \log \pi_{\text{ref}}(\mathbf{y} | \mathbf{x}) - Z(\mathbf{x}), \tag{6}$$

where $\mathbf{x}$ denotes the prompt and y denotes the response, $\pi^*$ is the optimal policy and $Z(\mathbf{x})$ is the partition function that normalizes it. Substituting eq. (6) into the Bradley Terry preference model, and leverage the maximum likelihood objective, DPO derives the loss:

$$\mathcal{L}_{\text{DPO}}(\pi_\theta; \pi_{\text{ref}}) = -\mathbb{E}_{(\mathbf{x}, \mathbf{y}^w, \mathbf{y}^l) \sim \mathcal{D}} \left[ \log \sigma \left( \beta \log \frac{\pi_\theta(\mathbf{y}^w | \mathbf{x})}{\pi_{\text{ref}}(\mathbf{y}^w | \mathbf{x})} - \beta \log \frac{\pi_\theta(\mathbf{y}^l | \mathbf{x})}{\pi_{\text{ref}}(\mathbf{y}^l | \mathbf{x})} \right) \right], \tag{7}$$

where $\sigma$ denotes the logistic function, $\mathbf{y}^w$ and $\mathbf{y}^l$ denote the preferred and dis-preferred responses to the prompt $\mathbf{x}$.

### 2.2.2 Deriving the Step-APO Objective

In the general maximum entropy RL setting (Ziebart, 2010), the optimal policy $\pi^*(\mathbf{a}|\mathbf{s})$ of multi-step RL objective in eq. (5) is:

$$\pi^*(\mathbf{a}_t|\mathbf{s}_t) = e^{(Q^*(\mathbf{s}_t,\mathbf{a}_t)-V^*(\mathbf{s}_t))/\beta}, \tag{8}$$

where $Q^*(\mathbf{s},\mathbf{a})$ is the optimal Q-function which models the expected future reward from $(\mathbf{s}_t, \mathbf{a}_t)$ under $\pi^*$. The optimal value function $V^*$ estimates the expected future reward under state $\mathbf{s}_t$, and it's a function of $Q^*$ (Rafailov et al., 2024).

Under the reward $r$ with a KL divergence penalty, the relationship between Q-function and step-level reward function can be established with the Bellman equation as follows:

$$Q^*(\mathbf{s}_t, \mathbf{a}_t) = r(\mathbf{s}_t, \mathbf{a}_t) + \beta \log \pi_{\text{ref}}(\mathbf{a}_t|\mathbf{s}_t) + V^*(\mathbf{s}_{t+1}). \tag{9}$$

By log-linearizing the optimal policy in eq. (8) and substituting in the Bellman equation from eq. (9) (Nachum et al., 2017; Rafailov et al., 2024), we have below equation which is precisely the optimal advantage function $A^*(\mathbf{s},\mathbf{a}) = Q^*(\mathbf{s},\mathbf{a}) - V^*(\mathbf{s})$:

$$\beta \log \frac{\pi^*(\mathbf{a}_t|\mathbf{s}_t)}{\pi_{\text{ref}}(\mathbf{a}_t|\mathbf{s}_t)} = r(\mathbf{s}_t, \mathbf{a}_t) + V^*(\mathbf{s}_{t+1}) - V^*(\mathbf{s}_t). \tag{10}$$

Unlike DPO utilize response-level Bradley Terry model, we introduce step-level Bradley Terry preference model to learn fine-grained step-level preference:

$$p^*(\mathbf{a}^w \succeq \mathbf{a}^l|\mathbf{s}) = \frac{\exp\left(r(\mathbf{s}, \mathbf{a}^w)\right)}{\exp\left(r(\mathbf{s}, \mathbf{a}^w)\right) + \exp\left(r(\mathbf{s}, \mathbf{a}^l)\right)}. \tag{11}$$

By substituting eq. (10) into eq. (11) and leveraging the negative log-likelihood loss, we derive the objective for Step-APO:

$$
\begin{aligned}
\mathcal{L}_{\text{Step-APO}}(\pi_\theta; \pi_{\text{ref}}) = -\mathbb{E}_{(\mathbf{s}_t,\mathbf{a}_t^w,\mathbf{a}_t^l)\sim\mathcal{D}} &\left[ \log \sigma \left( \beta \log \frac{\pi_\theta(\mathbf{a}_t^w \mid \mathbf{s}_t)}{\pi_{\text{ref}}(\mathbf{a}_t^w \mid \mathbf{s}_t)} + V(\mathbf{s}_t) - V(\mathbf{s}_{t+1}^w) \right.\right. \\
&\left.\left. - \left( \beta \log \frac{\pi_\theta(\mathbf{a}_t^l \mid \mathbf{s}_t)}{\pi_{\text{ref}}(\mathbf{a}_t^l \mid \mathbf{s}_t)} + V(\mathbf{s}_t) - V(\mathbf{s}_{t+1}^l) \right) \right) \right] \\
= -\mathbb{E}_{(\mathbf{s}_t,\mathbf{a}_t^w,\mathbf{a}_t^l)\sim\mathcal{D}} &\left[ \log \sigma \left( \beta \log \frac{\pi_\theta(\mathbf{a}_t^w \mid \mathbf{s}_t)}{\pi_{\text{ref}}(\mathbf{a}_t^w \mid \mathbf{s}_t)} - V(\mathbf{s}_{t+1}^w) \right.\right. \\
&\left.\left. - \beta \log \frac{\pi_\theta(\mathbf{a}_t^l \mid \mathbf{s}_t)}{\pi_{\text{ref}}(\mathbf{a}_t^l \mid \mathbf{s}_t)} + V(\mathbf{s}_{t+1}^l) \right) \right].
\end{aligned}
\tag{12}
$$

where $V(\mathbf{s}_{t+1}^w) - V(\mathbf{s}_{t+1}^l)$ denotes the advantage of $\mathbf{s}_{t+1}^w$ to $\mathbf{s}_{t+1}^l$ from the same start state.

To understand the difference between our Step-APO and other step-level DPO, we will analyze the gradient of the $\mathcal{L}_{\text{Step-APO}}$:

$$
\begin{aligned}
\nabla_\theta \mathcal{L}_{\text{Step-APO}}(\pi_\theta; \pi_{\text{ref}}) = -\beta \mathbb{E}_{(\mathbf{s}_t,\mathbf{a}_t^w,\mathbf{a}_t^l)\sim\mathcal{D}} &\left[ \sigma\left( \hat{r}_\theta(\mathbf{s}_t, \mathbf{a}_t^l) - \hat{r}_\theta(\mathbf{s}_t, \mathbf{a}_t^w) \right.\right. \\
&\left.\left. + V(\mathbf{s}_{t+1}^w) - V(\mathbf{s}_{t+1}^l) \right) \right] \left[ \nabla_\theta \log \pi(\mathbf{a}_t^w \mid \mathbf{s}_t) - \nabla_\theta \log \pi(\mathbf{a}_t^l \mid \mathbf{s}_t) \right].
\end{aligned}
\tag{13}
$$

where $\hat{r}_\theta(\mathbf{s}_t, \mathbf{a}_t) = \beta \log \frac{\pi_\theta(\mathbf{a}_t|\mathbf{s}_t)}{\pi_{\text{ref}}(\mathbf{a}_t|\mathbf{s}_t)}$. Intuitively, the gradient of the loss function $\mathcal{L}_{\text{Step-APO}}$ increases the likelihood of the preferred completions $\mathbf{a}_t^w$ and decreases the likelihood of dispreferred completions $\mathbf{a}_t^l$. Importantly, besides the examples are weighed by how much higher the $\hat{r}_\theta$ incorrectly orders the completions, the examples are also weighted by how much higher the advantage of $\mathbf{a}_t^w$ is compared to $\mathbf{a}_t^l$. This allows for assigning different optimization weights and emphasizes critical steps. Our experiments prove the importance of this weighting.

### 2.3 Iterative Training of Policy and Value Model

Our approach employs iterative training for policy and value models. Our policy model $\pi_\theta$ and value model $v_\phi$ are two separate models, both adapted from the same base model. We add a value head for the value model, which is randomly initialized in the first round. However, as the MCTS simulations proceed in the first round, rewards from terminal nodes are back-propagated to the intermediate nodes, reducing the negative impact of the random value initialization.

For policy model training, we first supervised fine-tune (SFT) it using collected correct paths from MCTS, then apply our Step-APO (eq. (12)) using step-level preference data also collected from MCTS. Notably, $V(\mathbf{s}_{t+1}^w)$ and $V(\mathbf{s}_{t+1}^l)$ in eq. (12), obtained from MCTS, represent the values of the corresponding states. The difference between these values reflects the advantage difference of the two actions under the same previous state $\mathbf{s}_t$:

$$A(\mathbf{s}_t, \mathbf{a}_t^w) - A(\mathbf{s}_t, \mathbf{a}_t^l) = Q(\mathbf{s}_t, \mathbf{a}_t^w) - V(\mathbf{s}_t) - (Q(\mathbf{s}_t, \mathbf{a}_t^l) - V(\mathbf{s}_t)) = V(\mathbf{s}_{t+1}^w) - V(\mathbf{s}_{t+1}^l). \quad (14)$$

For value model optimization, we use a mean squared error (MSE) loss between the value model's predict and values from MCTS. With the updated policy and value models, we can advance to the next-round MCTS, iterating this training process to enhance the models.

## 3 Experiments

### 3.1 Implementation Details

We iteratively generate data through MCTS and train our policy and value models in two rounds. In each round, the policy model generates plan steps and final solution steps by employing MCTS to address the given problem. The value model is utilized to assist in evaluating the intermediate steps during MCTS. At the end of each round, the generated data is used to train both the policy model and the value model.

**Model Architecture** We employ the DeepSeekMathBase-7B (Shao et al., 2024) as our initial policy model and add a randomly initialized value head to this model, serving as the initial value model. We then optimize these two distinct models independently and utilize the updated models for the next round of data generation.

**Datasets** We construct our training data using the training sets of GSM8K (Cobbe et al., 2021) and MATH (Hendrycks et al., 2021b) datasets. GSM8K comprises 7,473 training and 1,319 test problems, while MATH includes 7,500 training and 5,000 test problems. From these datasets, we exclusively extracted question-answer pairs from the training sets of GSM8K and MATH, omitting the human-annotated solution. This resulted in a total of 15k question-answer pairs for our training data.

**Training Data Generation via MCTS** For each problem, we employ MCTS to generate multiple step-level plans and final solutions. During the MCTS expansion phase, we expanded 5 child nodes for the root node and 3 child nodes for other nodes. The search is conducted with a maximum depth of 6. We apply a temperature of 0.7 to encourage diverse generation.

In the first round, we generate data from a subset of 5k question-answer pairs, consisting of 4k from the MATH and 1k from GSM8K, for efficiency. We carefully design prompts and 2-shot demonstrations to guide the model's output, see Appendix A for details. We perform a large number of MCTS simulations, specifically 200, in this phase to mitigate the impact of the random initialization of the value model. Starting from the second round, with the fine-tuned models from the first round, we utilize the full set of 15k question-answer pairs for data generation. A 2-shot prompt formatted in XML is used, and we perform 100 MCTS simulations.

**Training Data Construction** We utilize the state value $V$ of each node in MCTS to construct preference data. For plan step preference data, we categorize sibling nodes as "preferred" if their value is greater than 0 and "dispreferred" if their value is less than 0, forming preference pairs from any combination. For the final solution step data, we randomly select one preference pair for each parent node to construct the dataset. This is based on our

experimental findings that an excess of solution data can negatively impact the performance on out-of-domain reasoning tasks, whereas increasing the emphasis on plan data improves performance in both mathematical and other reasoning tasks (see subsection 3.5). We list the statistics for the generated data in two rounds in Appendix B.

**Training Details** For the policy model, we first randomly select up to four correct responses per problem for supervised fine-tuning (SFT). Next, we employ step-level preference data from MCTS to train the model with our Step-APO algorithm. For the value model, we use state value $V$ from MCTS for partial responses as labels to update the model. This allows the value model to score both partial plans and complete responses. The training hyperparameters are provided in Appendix D. Notably, because the value difference for final solution step preference pairs is 2, while the average value difference for other plan steps ranges between 0.6 and 0.8, we apply a scaling factor of 0.3 to the values of solution steps in Step-APO. In the second round of training, we utilize the data from the second round to train the base model, rather than the Round 1 model.

## 3.2 Main Results

Table 1: Main results on in-domain and out-of-domain reasoning tasks. Baseline results on MATH and GSM8K are reproduced. * denotes results from Self-Explore-GSM8K. - indicates that the model uses Python code interpreter and is not comparable with our method. Best results are bolded.

| Model | In domain | | Out-of-Domain | | | | |
|---|---|---|---|---|---|---|---|
| | MATH | GSM8K | HumanEval | ARC-C | GQPA | BBH | MMLU-stem |
| DeepseekMath-Base | 35.18 | 63.23 | 40.90 | 52.05 | 25.75 | 58.79 | 52.74 |
| STaR | 37.68 | 70.13 | 43.29 | 52.73 | 27.78 | 60.45 | 54.20 |
| Self-Explore-MATH | 37.86 | **78.39**\* | 41.46 | 54.01 | 33.83 | 60.04 | 54.04 |
| AlphaMath | - | - | 49.39 | 53.41 | 33.33 | 56.63 | 55.31 |
| CPL(Round1 SFT) | 36.30 | 63.79 | 42.68 | 54.44 | 28.78 | 59.68 | 54.58 |
| CPL(Round1 Step-APO) | 40.56 | 71.06 | 46.34 | 55.55 | 31.31 | 60.18 | 55.15 |
| CPL(Round2 SFT) | 39.16 | 69.75 | 48.78 | 54.95 | 29.79 | 59.93 | **55.44** |
| CPL(Round2 Step-APO) | **41.64** | 73.77 | **53.05** | **56.06** | **34.34** | **60.54** | 54.93 |

We evaluate our method on both mathematical tasks and out-of-domain reasoning tasks, as shown in Table 1. Evaluation details are provided in the Appendix C.

**Baseline** Our baseline includes DeepseekMath-Base-7B (Shao et al., 2024), along with three additional baselines, STaR (Zelikman et al., 2022), Self-Explore (Hwang et al., 2024), AlphaMath (Chen et al., 2024). We reran STaR by repeated sampling task-specific solutions using the same model and data as ours. The latter two baselines are developed based on DeepSeekMath-Base and utilize solution-centric search methods, employing the GSM8K and MATH datasets, distinguishing them from our proposed plan-based search methodology.

**Mathematical tasks** We evaluate CPL in-domain capabilities on MATH (Hendrycks et al., 2021b) and GSM8K (Cobbe et al., 2021).

- **MATH**: Comprising 5,000 intricate competition-level problems, aimed at evaluating the models' capability to perform complex mathematical reasoning.
- **GSM8K**: Containing 1,320 diverse grade school math problems, designed to assess basic arithmetic and reasoning skills in an educational context.

On in-domain tasks, CPL demonstrated significant performance improvements over DeepseekMath-Base on both the MATH and GSM8K datasets. Compared to Self-Explore, which leverages human-annotated rational data and extensive self-generated data for fine-tuning, CPL does not use any human-annotated rational data and relies solely on the model's self-exploration with much less SFT data. While Self-Explore performs better on simpler tasks like GSM8K, possibly due to the use of golden rationales and more SFT data, our method significantly outperforms it on more challenging tasks like MATH, which require more complex self-exploration of reasoning paths. These results underscore the efficacy of plan-based learning in enhancing the model's in-domain reasoning capabilities.

In both rounds, Step-APO consistently improves results over the SFT. Additionally, Round 2 outperforms Round 1 in both SFT and Step-APO, demonstrating that the updated policy and value models generate better data through MCTS, further improving performance.

**Out-of-domain reasoning tasks** We select five benchmarks for evaluating out-of-domain reasoning: HumanEval (Chen et al., 2021), ARC-C (Clark et al., 2018), GPQA (Rein et al., 2023), BBH (Suzgun et al., 2022), and MMLU-STEM (Hendrycks et al., 2021a). We employ few-shot prompting to evaluate these benchmarks.

- **HumanEval**: HumanEval is a widely used benchmark for code generation tasks. It provides descriptive prompts for each problem, prompting LLMs to generate corresponding code. It contains 164 problems.

- **ARC-C**: ARC includes questions derived from various grade-level science exams, designed to test models' ability to handle both straightforward and complex scientific queries. The challenge subset contains 1,172 test questions.

- **GPQA**: Providing "Google-proof" questions in the fields of biology, physics, and chemistry, GPQA is designed to test deep domain expertise and reasoning under challenging conditions. We use the diamond subset, which contains 198 difficult problems.

- **BIG-Bench Hard (BBH)**: Comprising 23 tasks previously identified as challenging for language models in the BIG-Bench benchmark, BBH contains a total of 6,511 challenging problems, aimed at evaluating the capabilities of large language models (LLMs) in solving these tasks.

- **MMLU-STEM**: Spanning 57 subjects across multiple disciplines, MMLU evaluates the breadth and depth of a model's knowledge in a manner similar to academic and professional testing environments. We select the STEM subset, which contains 3,130 problems.

As shown in Table 1, CPL also achieves significant improvements on OOD tasks, with average improvements of 5.7%, 3.1%, and 2.2% compared to the base model, Self-Explore-MATH and AlphaMath, respectively. This demonstrates that CPL enhances the model's generalization ability across diverse reasoning tasks. Compared to AlphaMath, which was trained on the same 15k dataset for 3 rounds, our performance on these OOD reasoning tasks is much better. Notably, AlphaMath even shows a decrease in performance on certain tasks, such as a 2.2% drop in BBH.

Unlike baseline methods that focus on task-specific solutions within the vast reasoning action space of LLMs, CPL concentrates on exploring the action space on high-level abstract plans. These plans embody abstract problem-solving strategies, enabling models to develop broader, task-agnostic abilities that enhance generalization.

### 3.3 ADVANTAGE OF PLAN-BASED LEARNING

In our early experiments, we aimed to validate whether plan-based learning offers superior generalization in reasoning tasks compared to solution-based learning. To this end, we utilized the GSM8K and MATH training sets to train our models and evaluated their performance on the BBH dataset.

Specifically, we compared two fine-tuning approaches: one using solution-based chain-of-thought (CoT) (Wei et al., 2023) formatted data and the other using plan-based formatted data. Both methods fine-tuned the model using the responses generated by the model itself, with each question generating only one response, and the data filtered based on the correctness of the answers. The results, as shown in Table 2, indicate

Table 2: Advantage of Plan-based Learning

| Model | BBH |
|---|---|
| DeepSeekMath-Base | 58.79 |
| Solution-based SFT | 58.92 |
| Plan-based SFT | **59.50** |

that plan-based learning enhances performance on the BBH dataset, while the solution-based approach does not demonstrate significant improvements. This finding is further supported

by the results in Table 1, where plan-based learning CPL consistently outperforms the solution-based learning baseline on out-of-domain tasks. Together, these results clearly demonstrate the advantage of plan-based learning.

## 3.4 ADVANTAGE OF STEP-APO

Table 3: Advantage of Step-APO

| Model | In domain | | Out-of-Domain | | | | |
|---|---|---|---|---|---|---|---|
| | MATH | GSM8K | HumanEval | ARC-C | GQPA | BBH | MMLU-stem |
| SFT | 36.30 | 63.79 | 42.68 | 54.44 | 28.78 | 59.68 | 54.58 |
| Instance-DPO | 37.72 | 69.29 | 43.90 | 54.61 | 24.24 | 60.13 | 54.42 |
| TDPO | 39.12 | 69.90 | 48.78 | 54.61 | 28.29 | 59.94 | 54.08 |
| Step-DPO | 37.89 | 69.83 | 42.68 | 54.44 | 25.25 | 59.44 | 54.68 |
| Step-APO | **40.56** | **71.06** | **48.78** | **55.55** | **31.31** | **60.18** | **55.15** |

To investigate the advantage of Step-APO, we compared the performance of Instance-DPO, TDPO (Zeng et al., 2024), Step-DPO, and Step-APO using data obtained from the first round of MCTS. Instance-DPO involves preference learning on the model's complete response based on the correctness of the final answer. Step-DPO and Step-APO, on the other hand, performs finer-grained learning on each step within the model's response. The difference is that Step-APO incorporates advantage estimates for preference pairs obtained through MCTS, while Step-DPO does not. TDPO improves DPO by extending it to the token level and incorporates forward KL divergence constraints for each token, improving alignment. All four preference learning strategies were applied to the model after supervised fine-tuning (SFT). The experimental results are summarized in Table 3.

The results show that all four preference learning methods achieve performance improvements over SFT in in-domain tasks. Step-DPO, due to its finer-grained supervision signal, slightly outperforms Instance-DPO. TDPO shows notable performance improvements, highlighting its effectiveness. However, its performance is inferior to our Step-APO across all tasks. On OOD tasks, Instance-DPO, TDPO and Step-DPO exhibit suboptimal performance. We believe this may be due to their failure to capture critical plan steps, thereby failing to enhance generalization. In contrast, our Step-APO algorithm consistently demonstrates performance improvements on OOD tasks.

## 3.5 DATA CONSTRUCTION

To investigate how to construct step-level preference data, we use MATH as a representative in-domain task and BBH and GPQA as representative out-of-domain tasks to compare three methods.

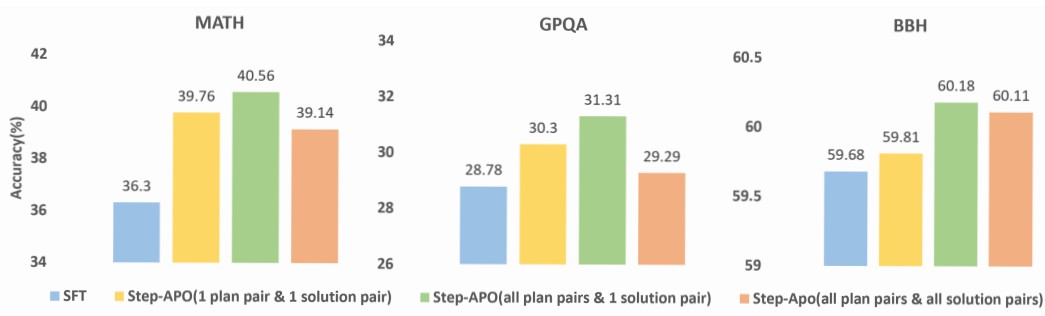

Figure 3: Impact of data construction. We outline different strategies for constructing Step-APO preference data based on the value $V$ of each node in MCTS: 1 plan pair (selecting the plan step with the maximum positive $V$ and the plan with the minimum negative $V$), 1 solution pair (randomly selecting one correct and one incorrect solution step), all plan pairs (selecting all combinations of plan steps with positive and negative $V$), and all solution pairs (selecting all combinations of correct and incorrect solution steps).

Initially, we construct preference data by creating at most one pair for all sibling nodes: for plan steps, we select the plan with the highest positive value and the plan with the lowest negative value; for solution steps, we randomly select one correct and one incorrect solution. Using Step-APO on this data, we observe performance improvements over the SFT model in both in-domain and out-of-domain reasoning tasks. Next, we enhance the plan step data by selecting all combinations of plans with positive and negative values while keeping the solution step data unchanged, which leads to further performance gains across both task types. However, continuously expanding the solution step data results in decreased model performance. Ultimately, we adopt the strategies of using all plan pairs and one solution pair for our experiments.

## 4 Related Work

**Search-Guided Reasoning in LLMs** Recent advancements (Feng et al., 2023; Chen et al., 2024; Xie et al., 2024) in enhancing LLM reasoning capabilities have focused on integrating Monte Carlo Tree Search (MCTS) to collect trajectories and train models, resulting in notable improvements in reasoning tasks. MCTS strikes a balance between exploration and exploitation, utilizing its look-ahead ability to obtain high-quality step-level supervision. For example, AlphaMath (Chen et al., 2024) employs MCTS to automatically generate process supervision, leading to significant improvements in mathematical reasoning. However, these MCTS-based training methods face challenges such as vast search spaces, limited solution diversity for LLMs. Furthermore, there is limited research on how these methods generalize to other reasoning tasks and enhance overall reasoning capabilities. To address these issues, we propose a method for searching over plan steps and learning critical plan steps for problem-solving, which aims to enhance generalization in reasoning tasks.

**Direct Preference Optimization (DPO) Algorithms** DPO (Rafailov et al., 2023) uses instance-level preference data for model optimization but has notable limitations. It struggles with multi-step reasoning tasks because it cannot effectively correct specific errors within the reasoning process (Hwang et al., 2024). Moreover, training on model-generated positive data can amplify spurious correlations from incorrect intermediate steps, leading to poor generalization (Setlur et al., 2024). Recent work proposes step-level DPO (Setlur et al., 2024; Lai et al., 2024) to address these issues by providing the fine-grained error identification needed for improving reasoning capabilities. For example, Self-Explore (Hwang et al., 2024) identifies the first incorrect step in a solution and constructs step-level preference data to guide model improvement. Unlike these heuristic methods, we propose Step-APO to fully explore the step-level search space and achieve the maximum optimization potential.

## 5 Conclusion

Scaling RL to develop a general reasoner remains an open and important research question. In this work, we explore the scaling problem in RL and propose searching within the action space on high-level abstract plans to enhance model generalization, rather than focusing on task-specific action spaces that often limit generalization. Additionally, we introduce CPL, which uses plan-based search and finer-grained learning of plan step preferences to enable the model to identify critical plan steps within the reasoning trace, thereby enhancing its overall reasoning ability. Ultimately, CPL successfully improves transfer performance in various out-of-domain tasks and offers valuable contributions to future research in RL scaling.

In future work, we will expand our current plan strategy beyond the option to continue to the next reasoning step. We will consider more diverse planning strategies, such as self-correction and the exploration of new ideas to solve problems. Additionally, test-time search has been shown to significantly enhance the model's reasoning capabilities on complex problems. We could combine test-time compute with our training-time compute. Utilizing our value model for test-time search has the potential to further improve the model's reasoning performance, and we will explore this in future work. Furthermore, test-time search in a vast action space poses high latency issues. We will continue to investigate how to effectively learn critical steps for problem-solving to enable efficient search.

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

## A  PROMPT USED IN MCTS

Prompts for Round 1 and Round 2 are listed below.

702
703
704
705
706
707
708
709
710
711
712
713
714
715
716
717
718
719
720
721
722
723
724
725
726
727
728
729
730
731
732
733
734
735
736
737
738
739
740
741
742
743
744
745
746
747
748
749
750
751
752
753
754
755

### Round 1 2-shot prompt

```
You are a powerful agent with advanced reasoning and planning
capabilities. Answer the questions as best you can.

!!!Remember:
1. Your answer should have two sections: "Plans" and "Detailed
Implementation".
2. In the "Plans" section, you should outline step-by-step plans for
solving the problem. These plans might include extracting key
information, forming sub-questions, analyzing aspects, etc. Each step
should introduce new insights, avoid overly abstract or generic
actions. End each step with "<endstep>".
3. In the "Detailed Implementation" section, provide detailed steps
that correspond to each plan, and conclude with "The final answer is
\boxed{answer}.<endsolution>"

The following is a template for your answer:

Question: The input question

Plans:
Plan 1: Describe the first plan step.<endstep>
Plan 2: Describe the second plan step<endstep>
...
Plan N: Describe the final plan step<endstep>

Detailed Implementation:
1. Execute the first plan step
2. Execute the second plan step
...
N. Execute the final plan step
The final answer is \boxed{answer}.<endsolution>

The following are 2 demonstration examples.

Question: Natalia sold clips to 48 of her friends in April, and then
she sold half as many clips in May. How many clips did Natalia sell
altogether in April and May?

Plans:
Plan 1: Analyze the total number of clips sold in April.<endstep>
Plan 2: Calculate the number of clips sold in May by applying the
"half as many" condition to the number sold in April.<endstep>
Plan 3: Sum the results from April and May to determine the overall
total of clips sold over the two months.<endstep>

Detailed Implementation:
1. Natalia sold 48 clips in April.
2. The number of clips sold in May is $\frac{48}{2}=24$.
3. The total number of clips sold in April and May combined is
$48+24=72$.
The final answer is \boxed{72}.<endsolution>

Question: If $x^2+y^2=1$, what is the largest possible value of
$|x|+|y|$?
```

Plans:
Plan 1: Understand that the equation $x^2+y^2=1$ defines a circle centered at the origin with a radius of 1. To maximize $|x|+|y|$, we need to consider points on this circle that maximize the sum of the absolute values of $x$ and $y$.<endstep>
Plan 2: Recognize that $|x|+|y|$ is maximized when both $|x|$ and $|y|$ are large. The maximum sum occurs along lines where $x$ and $y$ contribute equally, specifically along the lines $y=x$ and $y=-x$.<endstep>
Plan 3: Identify the points of intersection between the lines $y=x$ and $y=-x$ with the circle $x^2+y^2=1$. These points are expected to yield the maximum value of $|x|+|y|$.<endstep>
Plan 4: Evaluate $|x|+|y|$ for the intersection points to determine the maximum possible value.<endstep>

Detailed Implementation:
1. The circle $x^2+y^2=1$ is centered at the origin with a radius of 1. We need to find the points on this circle that maximize the sum $|x|+|y|$.
2. To maximize $|x|+|y|$, the sum is largest when both $|x|$ and $|y|$ are large. This occurs along the lines $y=x$ and $y=-x$, where $x$ and $y$ contribute equally to the sum.
3. The intersection points are $\left(\frac{1}{\sqrt{2}},\frac{1}{\sqrt{2}}\right)$, $\left(\frac{1}{\sqrt{2}},-\frac{1}{\sqrt{2}}\right)$, $\left(-\frac{1}{\sqrt{2}},\frac{1}{\sqrt{2}}\right)$, and $\left(-\frac{1}{\sqrt{2}},-\frac{1}{\sqrt{2}}\right)$.
4. For these points, calculate $|x|+|y|$. For $\left(\frac{1}{\sqrt{2}},\frac{1}{\sqrt{2}}\right)$, $|x|+|y|=\sqrt{2}$. The same value applies to the other points. Therefore, the maximum value is $\sqrt{2}$.
The final answer is $\boxed{\sqrt{2}}$.<endsolution>

Now! It's your turn.

---

**Round 2 XML 2-shot prompt**

```
<question>
Question: Natalia sold clips to 48 of her friends in April, and then
she sold half as many clips in May. How many clips did Natalia sell
altogether in April and May?
</question>
<plan>
<step>
Plan 1: Analyze the total number of clips sold in April.
</step>
<step>
Plan 2: Calculate the number of clips sold in May by applying the
"half as many" condition to the number sold in April.
</step>
<step>
Plan 3: Sum the results from April and May to determine the overall
total of clips sold over the two months.
</step>
</plan>
<solution>
```

```
1. Natalia sold 48 clips in April.
2. The number of clips sold in May is $\frac{48}{2}=24$.
3. The total number of clips sold in April and May combined is
$48+24=72$.
The final answer is \boxed{72}.
</solution>

<question>
If $x^2+y^2=1$, what is the largest possible value of $|x|+|y|$?
</question>
<plan>
<step>
Plan 1: Understand that the equation $x^2+y^2=1$ defines a circle
centered at the origin with a radius of 1. To maximize $|x|+|y|$, we
need to consider points on this circle that maximize the sum of the
absolute values of $x$ and $y$.
</step>
<step>
Plan 2: Recognize that $|x|+|y|$ is maximized when both $|x|$ and
$|y|$ are large. The maximum sum occurs along lines where $x$ and $y$
contribute equally, specifically along the lines $y=x$ and $y=-x$.
</step>
<step>
Plan 3: Identify the points of intersection between the lines $y=x$
and $y=-x$ with the circle $x^2+y^2=1$. These points are expected to
yield the maximum value of $|x|+|y|$.
</step>
<step>
Plan 4: Evaluate $|x|+|y|$ for the intersection points to determine
the maximum possible value.
</step>
</plan>
<solution>
1. The circle $x^2+y^2=1$ is centered at the origin with a radius of
1. We need to find the points on this circle that maximize the sum
$|x|+|y|$.
2. To maximize $|x|+|y|$, the sum is largest when both $|x|$ and
$|y|$ are large. This occurs along the lines $y=x$ and $y=-x$, where
$x$ and $y$ contribute equally to the sum.
3. The intersection points are
$\left(\frac{1}{\sqrt{2}},\frac{1}{\sqrt{2}}\right)$,
$\left(\frac{1}{\sqrt{2}},-\frac{1}{\sqrt{2}}\right)$,
$\left(-\frac{1}{\sqrt{2}},\frac{1}{\sqrt{2}}\right)$, and
$\left(-\frac{1}{\sqrt{2}},-\frac{1}{\sqrt{2}}\right)$.
4. For these points, calculate $|x|+|y|$. For
$\left(\frac{1}{\sqrt{2}},\frac{1}{\sqrt{2}}\right)$,
$|x|+|y|=\sqrt{2}$. The same value applies to the other points.
Therefore, the maximum value is $\sqrt{2}$.
The final answer is $\boxed{\sqrt{2}}$.
</solution>
```

## B  Statistic for the generated data

We list the statistic for the generated data in two rounds in Table 4. Round 2 generates more correct responses, indicating a stronger policy and value model.

Table 4: Statistic for the generated data in two rounds

| Round Num | Avg Depths | Pos:Neg | Plan Pairs Count | Solution Pairs Count |
|---|---|---|---|---|
| Round 1 | 4.18 | 1:3.16 | 18742 | 16506 |
| Round 2 | 3.80 | 1:1.23 | 24707 | 24633 |

## C    EVALUATION DETAILS

**Mathematical Tasks** For our CPL, we evaluated in-domain reasoning capabilities using a zero-shot setting on the MATH and GSM8K datasets. We utilized vLLM (Kwon et al., 2023) for inference during evaluation and employed the math evaluation toolkit (Zhang et al., 2024) to assess model-generated answers. For other baseline models, results were reproduced on our machine using the configurations and codes from the original papers.

**Out-of-domain Reasoning Tasks** For all models, we employed few-shot prompting through the lm-evaluation-harness (Gao et al., 2024) to evaluate performance on ARC-C (25-shot), BBH (3-shot), and MMLU-stem (5-shot). Following Yue et al. (2024), we utilized 5-shot prompting to evaluate the GPQA diamond subset. Following Chen et al. (2021), we utilized zero-shot setting to evaluate performance on HumanEval.

## D    IMPLEMENTATION DETAILS

Table 5: Key Hyperparameters of CPL

| Hyperparameter | Value |
|---|---|
| $c_{puct}$ | 1.5 |
| Simulations $N$ | 200 (for round 1) or 100 |
| Expand child nodes | 5 (for root) or 3 |
| Temperature | 0.7 |
| Max depth | 6 |
| SFT batch size | 512 |
| SFT learning rate | 1e-5 |
| SFT epochs | 5 (for round 1) or 3 |
| Step-APO batch size | 64 |
| Step-APO $\beta$ | 0.3 |
| Step-APO learning rate | 1e-6 |
| Step-APO epochs | 2 |
| Solution step scaling factor | 0.3 |
| Lr scheduler type | cosine |
| Warmup ratio | 0.1 |

All models in our experiments were trained on 8 * NVIDIA H100 GPUs. We implement our Step-APO in Llama Factory (Zheng et al., 2024) and use Llama Factory as the training framwork. We use vLLM (Kwon et al., 2023) as the inference framework. We train all models with DeepSpeed ZeRO Stage2 (Rajbhandari et al., 2021), Flash Attention 2 (Dao, 2023). The key hyperparameter of CPL is listed in Table 5.

## E    MORE RESULTS ON LLAMA 3

To further demonstrate the effectiveness of CPL across different models, we conducted additional experiments on the Llama-3-8B model (Meta, 2024). The experimental setup is consistent with Round 1 in Table 1. Specifically, we used MCTS to generate data from a subset of 5,000 question-answer pairs, comprising 4,000 from the MATH dataset and 1,000

from GSM8K. We then optimized the Llama-3-8B model using SFT and Step-APO. The results are presented in Table 6.

Table 6: Additional Results on Llama-3-8B

| Model | In domain | | Out-of-Domain | | | | |
|---|---|---|---|---|---|---|---|
| | MATH | GSM8K | HumanEval | ARC-C | GQPA | BBH | MMLU-stem |
| Llama-3-8B | 18.16 | 49.17 | 37.80 | 57.94 | 27.27 | 62.92 | 55.82 |
| Llama-3-8B + CPL(Round1) | **20.24** | **53.90** | **40.24** | **60.24** | **34.34** | **64.08** | **56.83** |

The experimental results show that after just one round of optimization, the CPL method significantly improved the model's performance on both in-domain and out-of-domain tasks. This demonstrates the robustness and effectiveness of the CPL across different models.

## F  DETAILS OF PLAN-BASED MCTS

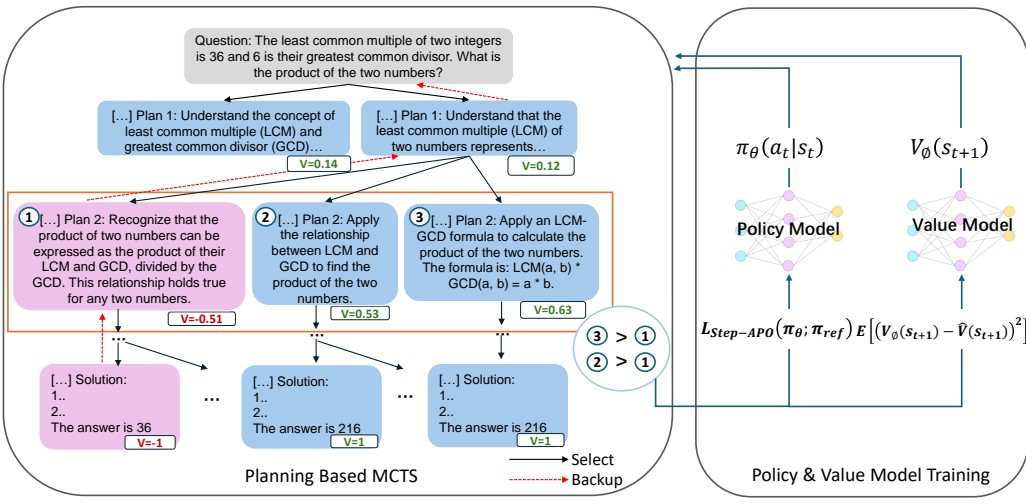

Figure 4: CPL boosts model performance via iterative process over planning based MCTS and step-level preference learning. **Left:** Example of an MCTS-generated plan tree, exploring diverse planning strategies in the vast search space. CPL generates step-by-step plans, which lead to the final solution and answer. State value $V$ is updated via a bottom-up reward propagation from the terminal node to the root, and used to assign preferences. **Right:** Step-level preferences from MCTS are used to update the policy and value models. Our Step-APO integrates value estimates for preference pairs into DPO, assigning different optimization weights to emphasize critical steps. The value model is optimized using MSE loss.

**Search on High-Level Abstract Plan.** We propose leveraging search on high-level abstract plans, expressed in natural language, as a universal interface across tasks. These plans represent abstract thinking for problem-solving, such as determining which knowledge to apply or how to decompose a problem, enabling models to develop broader, task-agnostic capabilities that enhance generalization (See Figure 2 Left).

**Integrating Plan into MCTS.** We design a carefully crafted two-shot prompt (See Appendix A) to guide the model in answering questions in two parts: (1) step-by-step plans, and (2) detailed implementation. In the MCTS search tree, each non-terminal node stores a single plan step, while terminal nodes store the detailed implementation for all preceding plan steps (See Figure 4 Left).

MCTS consists of the following phases:

- **Selection**: We use the PUCT algorithm to iteratively select nodes in the search tree, which leverages the LLMs' generation probability to guide the selection process, reducing the selection of unreasonable nodes.

- **Expansion**: In this phase, we expand selected node by using the current states as the prompt, which contain the question and all previously generated responses, to guide the model in generating the next plan step or the final detailed implementation.

- **Evaluation**: In this phase, we evaluate the value of newly expanded nodes. The evaluation process differs for non-terminal and terminal nodes:

  - For non-terminal nodes (plan steps), unlike traditional MCTS which typically uses simulations, we employ a value model to evaluate the node's value for efficiency.
  - For terminal nodes (detailed implementation), we parse the answer in the detailed implementation, compare it to the ground truth, and assign a value based on correctness.

- **Backup**: We backpropagate the value of newly expanded nodes to update the search tree.

**Optimizing the Policy and Value Models.** After each round of MCTS, as shown in Figure 4 Right, we optimize the policy model using SFT and the Step-APO algorithm. Additionally, we optimize the value model using Mean Squared Error (MSE) loss to ensure its outputs are closer to the state value $V$ of each node in the MCTS search tree.

## G   SUPPLEMENTARY DISCUSSION ON RELATED WORK

**Post-Training on Self-Generated Data for Reasoning Improvement** Table 7 summarizes the key distinctions between our approach and existing self-improvement methods in reasoning. These methods all improve reasoning ability through post-training on self-generated data, but differ in terms of search methods, supervision granularity, search space, and generalization capabilities across tasks.

Table 7: Key differences between existing self-improvement methods and our approach.

| Method | Search Method | Supervision | Search Space | Generalization |
|---|---|---|---|---|
| STaR (Zelikman et al., 2022) | Repeated sampling | Response-level | Task-specific solutions | ✗ |
| Self-Explore (Hwang et al., 2024) | Repeated sampling | Step-level | Task-specific solutions | ✗ |
| TS-LLM (Feng et al., 2023) | MCTS | Response-level | Task-specific solutions | ✗ |
| AlphaMath (Chen et al., 2024) | MCTS | Response-level | Task-specific solutions | ✗ |
| ALPHALLM (Tian et al., 2024) | MCTS | Response-level | Task-specific solutions | ✗ |
| MCTS-DPO (Xie et al., 2024) | MCTS | Step-level | Task-specific solutions | ✗ |
| CPL | MCTS | Step-level | Abstract plans | ✓ |

**Direct Preference Optimization (DPO) Algorithms** Recent work (Rafailov et al., 2024; Zeng et al., 2024; Zhong et al., 2024) provide a token-wise Markov decision process (MDP) formulation for Reinforcement Learning from Human Feedback (RLHF). Rafailov et al. (2024) derives DPO in a token-level MDP and demonstrates that DPO can be viewed as an inverse Q-learning algorithm. Zeng et al. (2024) improves DPO by extending it to the token level and incorporating forward KL divergence constraints for each token. Zhong et al. (2024) models RLHF problems as a token-wise MDP, introducing the RTO algorithm, which learns a token-level reward function from preference data and performs policy optimization based on this learned reward signal. Our Step-APO is based on step-level MDPs, which are better suited for capturing complex reasoning steps in reasoning tasks, whereas token-level MDPs focus more on each word choice during the generation process.

