# OpenReview forum: "CPL: Critical Plan Step Learning Boosts LLM Generalization in Reasoning Tasks"
_ICLR.cc/2025/Conference — ICLR 2025 Conference Withdrawn Submission_

### Official Review · Reviewer_gctS · 2024-10-30

**Soundness:** 3
**Presentation:** 2
**Contribution:** 3
**Rating:** 6
**Confidence:** 3

**Summary:**

This paper introduces Critical Plan Step Learning (CPL), a novel approach for enhancing the generalization of large language models (LLMs) in reasoning tasks by leveraging plan-based reinforcement learning (RL). CPL employs Monte Carlo Tree Search (MCTS) to explore high-level, abstract problem-solving strategies, distinguishing critical steps that improve reasoning. Experimental results show that CPL significantly boosts in-domain and out-of-domain task performance, underscoring its potential for broader LLM applications.

**Strengths:**

- Originality: The paper presents Critical Plan Step Learning (CPL), which focuses on identifying and learning critical steps in high-level abstract plans for generalization in reasoning tasks, a novel concept in reinforcement learning for LLMs.

- Quality: The paper includes clear figures and comprehensive experiments to illustrate the effectiveness of CPL.

- Significance: CPL demonstrates improvements over baseline models across both in-domain (e.g., GSM8K, MATH) and out-of-domain tasks (e.g., HumanEval, ARC-C).

- Clarity: The paper is well-organized and accessible, with clear descriptions of CPL’s components and step-by-step explanations of the methods, making complex concepts understandable.

**Weaknesses:**

- In the contributions and conclusion, you mentioned "critical steps", while critical steps in this paper is not well-defined theoretically or experimentally. This undermines the interpretability of the process and raises questions about the reproducibility and consistency of its critical steps across different tasks.
- The model label in Section 3.2 is unclear. What are CPL(Round2 SFT) and CPL-final respectively? I would think CPL(Round2 SFT) is the model after Round2 SFT but it seems like the performance of CPL(Round2 SFT) drops and is lower than the previous steps for some tasks.
- The iterative training with MCTS and Step-APO requires multiple rounds of data generation, which can be prohibitively resource-intensive. There is little discussion on scalability or the potential trade-offs in model efficiency, which are crucial for real-world deployment.

**Questions:**

1. As mentioned in Weakness, the performance of CPL(Round2 SFT) drops. If that denotes the model after Round2-SFT, could the authors provide insights into why the Round 2 SFT performance declines compared to Round 1?
2. GSM8K and MATH are both math-centric. Will the result be different if the in-domain tasks change? For example, HumanEval is used as in-domain tasks, and MATH and GSM8K as out-of-domain tasks.
3. Why was MCTS chosen over other search methods for exploring high-level plans? Were any alternative search techniques considered? Providing an ablation study about the search methods will be helpful.
4. Have you compared CPL with general SOTA LLMs, especially the ones with chain-of-thought/tree-of-thought prompting or related techniques?

---

> ### Author Response · Authors · 2024-11-20
> **Response to Reviewer gctS (1/3)**
>
> Thank you very much for your valuable feedback. We address your feedback point by point below.
>
> > Q1: In the contributions and conclusion, you mentioned "critical steps", while critical steps in this paper is not well-defined theoretically or experimentally. This undermines the interpretability of the process and raises questions about the reproducibility and consistency of its critical steps across different tasks.
>
> We define critical steps as steps that provide higher advantages over others, as illustrated in Figure 2 Right. Additionally, it has a clear theoretical definition for advantages. It is a general concept in RL and task-independent.
> - Eq.14 gives the definition of an action (i.e., a step) advantage and demonstrates that the advantage difference between two actions/steps equals the difference in state values.
> - Eq.13 shows that Step-APO assigns higher optimization weights to examples where the advantage difference between paired actions is large. This allows for emphasizing critical steps with higher advantages and decreasing errors in steps with low advantages.
>
> $A(s_t,a_t^w) - A(s_t,a_t^l) = Q(s_t,a_t^w) - V(s_t) -(Q(s_t,a_t^l) - V(s_t)) = V(s_{t+1}^w)-V(s_{t+1}^l) $ (Eq. 14)
> $ \nabla _ {\theta}\mathcal{L} _ \text{Step-APO}(\pi _ {\theta}; \pi _ {ref}) = - \beta \mathbb{E}_{(s_t, a_t^w, a_t^l) \sim \mathcal{D}} \bigg[ \sigma \Big( \hat{r} _ {\theta}(s_t, a_t^l) - \hat{r} _ {\theta} (s_t, a_t^w) + V(s _ {t+1}^w) - V(s _ {t+1}^l) \Big) \bigg] \bigg[\nabla _ {\theta} \log \pi(a_t^w \mid s_t) - \nabla _ {\theta} \log \pi(a_t^l \mid s_t) \bigg]$  (Eq. 13)
>
> > Q2: The model label in Section 3.2 is unclear. What are CPL(Round2 SFT) and CPL-final respectively? I would think CPL(Round2 SFT) is the model after Round2 SFT but it seems like the performance of CPL(Round2 SFT) drops and is lower than the previous steps for some tasks.
>
> > As mentioned in Weakness, the performance of CPL(Round2 SFT) drops. If that denotes the model after Round2-SFT, could the authors provide insights into why the Round 2 SFT performance declines compared to Round 1?
>
> We are pleased to provide further clarification.
>
> CPL(Round2 SFT) is the model after Round2 SFT, CPL-final is the model after Round2 Step-APO and we updated it in paper. We follow AlphaMath[2], where **Round 2 is also trained on the base model rather than being based on Round 1 Step-APO**. The improvement for the base model in Round 2 is much greater compared to Round 1, demonstrating that the updated policy and value models generate better data through MCTS, further improving performance.
>
> > Q3: The iterative training with MCTS and Step-APO requires multiple rounds of data generation, which can be prohibitively resource-intensive. There is little discussion on scalability or the potential trade-offs in model efficiency, which are crucial for real-world deployment
>
> - **For resource-intensive**. We believe that the computational resources required during the RL training phase are a worthwhile and manageable expense because training-based methods require these resources only once or periodically. Our method has also demonstrated that RL can significantly enhance the model's reasoning capabilities. Training with the CPL method on more data has the potential to develop a strong general reasoner.
> - **For model efficiency in real-world deployment**. Our CPL, as a training method, does not directly impact model efficiency. Our approach can be applied with or without test-time scaling. In our evaluation, we did not involve test-time scaling, meaning that there was no additional inference cost, yet the model's performance across a wide range of tasks was significantly improved.

---

> ### Author Response · Authors · 2024-11-20
> **Response to Reviewer gctS (2/3)**
>
> > Q4: GSM8K and MATH are both math-centric. Will the result be different if the in-domain tasks change? For example, HumanEval is used as in-domain tasks, and MATH and GSM8K as out-of-domain tasks.
>
> Thank you for your valuable suggestion, but there are significant differences between the MCTS implementation in code and the one in math. For example, in code, correctness evaluation is not as straightforward as in mathematics, where there are golden answers. Some work evaluates code quality using the pass rate of test cases, but such training data is scarce. Regarding datasets, the Humaneval dataset you mentioned contains only 164 test questions, without a provided training set. We are currently searching for other datasets that are suitable for training and include test cases. These differences make it challenging for us to implement MCTS for code in a short time. However, we are still working on this and doing our best to complete the experiment.
>
> On the other hand, because domains like code are difficult to obtain accurate RL supervisory signals (verifying code correctness can be costly), many current works using RL to improve reasoning focus on the math domain due to accurate and accessible supervision. This also highlights the value of our work. CPL is a promising step toward scaling RL across domains. It also addresses the critical challenge of RL supervision, **where obtaining accurate signals can be difficult**.  Using search-and-RL techniques with natural language plans on tasks containing high-quality data can generalize reasoning improvements to OOD tasks.
>
> > Q5: Why was MCTS chosen over other search methods for exploring high-level plans? Were any alternative search techniques considered? Providing an ablation study about the search methods will be helpful.
>
> Classic search methods include DFS, BFS, A*, and MCTS. Numerous approaches have demonstrated the success of MCTS in improving reasoning [1,2], and **AlphaZero [1] has conducted extensive experiments proving that MCTS significantly outperforms DFS/BFS methods on deep-search tasks** due to MCTS can efficiently explore complex search spaces. Therefore, we chose MCTS as our search method. It is worth noting that our approach is not limited to specific search methods. Below is a comparison of several search methods.
> - DFS/BFS are heuristic-based and can exhaustively explore the search space, while they ensure systematic exploration, their lack of prioritization mechanisms makes them inefficient for complex tasks, as they may spend significant time exploring unpromising branches.
> - A* uses heuristics to evaluate and navigate the search space. However, A* has notable limitations: it focuses solely on the optimal path according to the given heuristic, which means it does not explore alternative paths that may also hold valuable information. This lack of exploration can lead to a failure to collect diverse data for training the model.
> - MCTS uses simulations to balance exploration and exploitation, allowing it to focus on promising branches based on simulation outcomes. This approach allows MCTS to gather a broader range of data by sampling multiple potential solutions, enhancing its efficiency and diversity compared to both DFS/BFS and A*. This is extremely important for collecting training data to post-train LLMs.

---

> ### Author Response · Authors · 2024-11-20
> **Response to Reviewer gctS (3/3)**
>
> > Q6: Have you compared CPL with general SOTA LLMs, especially the ones with chain-of-thought/tree-of-thought prompting or related techniques?
>
> Thank you for your valuable suggestion.  Firstly, we want to emphasize that our approach is a general method that can be applied to any model.
>
> We compared DeepSeekMath  + CPL with Llama 3.1 8B-CoT:
> - Llama 3.1, as a SOTA general model, has gained strong general reasoning abilities through extensive training and performs very well on non-math reasoning tasks.
> - DeepSeekMath is specifically optimized for math tasks, rather than general reasoning tasks. When we apply CPL to train DeepSeekMath solely on math data, it can significantly improve its performance on OOD tasks, reaching levels that are close to, or even surpass, the performance of Llama 3.1. This demonstrates the strong generalization ability of CPL.
>
> | Model                | In-domain      |           |           | Out-of-domain    |           |           |           |
> |----------------------|----------------|-----------|-----------|------------------|-----------|-----------|-----------|
> |                      | MATH          | GSM8K     | HumanEval | ARC-C            | GPQA      | BBH       | MMLU-stem |
> | DeepseekMath-Base    | 35.18         |  60.23         | 40.90     | 52.05            | 25.75     | 58.79     | 52.74     |
> | Llama-3.1-8B-COT     | 20.56         | 47.58     | 28.66     | 57.59            | 25.75     | 63.40     | 55.98     |
> | DeepSeekMath  + CPL                 | 41.64         | 73.77     | 53.05     | 56.06            | 34.34     | 60.54     | 54.93     |
>
> ---
> **Reference**
> [1] Feng, Xidong, et al. "Alphazero-like tree-search can guide large language model decoding and training."  ICML 2024.
> [2] Chen, Guoxin, et al. "AlphaMath Almost Zero: process Supervision without process." NeurIPS 2024.
>
> &nbsp;
>
> We want to express our sincere gratitude for your review. If you have any further questions or concerns, please feel free to contact us at any time. We are always available and look forward to further discussions with you.
>
> Best regards,
> All Authors

---

> ### Comment · Reviewer_gctS · 2024-11-22
> **Official Comment by Reviewer gctS**
>
> Thank you for a clear response to my comments. With these additions and explanations, I have more understanding of the work and believe the work is strong enough. I have updated my score accordingly.

---

> > ### Author Response · Authors · 2024-11-24
> > **Thanks for your response and score adjustment**
> >
> > Dear Reviewer gctS,
> >
> > Thank you so much for taking the time to review our work and for acknowledging our response and efforts. We truly appreciate your thoughtful feedback and guidance.
> >
> > Sincerely,
> > All Authors

---

### Official Review · Reviewer_aihq · 2024-11-01

**Soundness:** 3
**Presentation:** 2
**Contribution:** 2
**Rating:** 5
**Confidence:** 3

**Summary:**

The paper introduces Critical Plan Step Learning (CPL), a novel approach to improve large language models' reasoning abilities through reinforcement learning. The method uniquely combines Monte Carlo Tree Search for exploring diverse reasoning strategies with Step-level Advantage Preference Optimization for learning critical planning steps. While trained only on mathematical datasets (GSM8K and MATH), CPL demonstrates significant improvements not only on these training domains but also generalizes well to various out-of-domain reasoning tasks, including programming, physics, and general STEM problems. This suggests CPL effectively enhances models' general reasoning capabilities rather than just task-specific performance.

**Strengths:**

- The paper proposes a novel plan-based MCTS simulation data construction method introduces a step-level APO algorithm
- Comprehensive experimental validation is conducted across various reasoning datasets
- The paper is well-written with clear structure and organization

**Weaknesses:**

- Using MCTS to sample high-quality data for fine-tuning is not a novel contribution, as [1] has already proposed a self-improvement method based on MCTS.
- The methodology description lacks clarity, particularly regarding how the plan is integrated into the MCTS process.
- The comparison with baselines is insufficient, as it lacks search-based algorithms such as [1][2] in the baseline comparisons.

[1]AlphaZero-Like Tree-Search can Guide Large Language Model Decoding and Training

[2]Toward Self-Improvement of LLMs via Imagination, Searching, and Criticizing

**Questions:**

- Could the authors provide more method details, particularly about how MCTS rewards are calculated and whether there is an additional reward model? Perhaps include a flowchart illustrating plan-based MCTS?
- Training-based methods consume substantial computational resources. Could you elaborate on the differences between training-based and testing-based approaches, as mentioned in [1][2]? If minimal resources are required at test time, is RL fine-tuning necessary?
- Please provide more results for different sampling methods in Section 3.3. The authors only compared with CoT methods - how does it compare with MCTS without planning, Tree of Thoughts, Q*[1], and other similar methods?

[1]:Q*: Improving Multi-step Reasoning for LLMs with Deliberative Planning

[2]:Scaling LLM Test-Time Compute Optimally can be More Effective than Scaling Model Parameters

---

> ### Author Response · Authors · 2024-11-20
> **Response to Reviewer aihq (1/3)**
>
> Thank you very much for your valuable feedback. We address your feedback point by point below.
> > Q1: Using MCTS to sample high-quality data for fine-tuning is not a novel contribution, as [1] has already proposed a self-improvement method based on MCTS.
>
> We want to emphasize that our core contribution tackles an important, yet less-explored problem: **how to better generalize reasoning abilities across tasks and develop a general reasoner.** The novelty of CPL lies in its distinction from existing RL-based reasoning methods, which are task-specific. **We propose using search (e.g., MCTS) and RL on natural language abstract plans as a universal interface across tasks,** enabling the model to generalize more effectively. CPL offers valuable insights and support for training general reasoners, which is of great significance, as outlined in the overall response.
>
> > Q2: The methodology description lacks clarity, particularly regarding how the plan is integrated into the MCTS process.
>
> > Could the authors provide more method details, particularly about how MCTS rewards are calculated and whether there is an additional reward model? Perhaps include a flowchart illustrating plan-based MCTS?
>
> Thank you for your valuable suggestion. We are pleased to provide further clarification to help in understanding our plan-based MCTS.
>
> **Integrating the Plan into MCTS**: We design a carefully crafted two-shot prompt (listed in `Appendix A`) to guide the model in answering questions in two parts: (1) step-by-step plans, and (2) detailed implementation. In the MCTS search tree, each non-terminal node stores a single plan step, while terminal nodes store the detailed implementation of all preceding plan steps.
>
> MCTS consists of the following phases:
> - **Selection**: We use the PUCT algorithm to iteratively select nodes in the search tree, which leverages the LLMs' generation probability to guide the selection process, reducing the selection of unreasonable nodes.
> - **Expansion**: In this phase, we expand selected node by using the current states as the prompt, which contain the question and all previously generated responses, to guide the model in generating the next plan step or the final detailed implementation.
> - **Evaluation**: In this phase, we evaluate the value of newly expanded nodes. The evaluation process differs for non-terminal and terminal nodes:
>    - For non-terminal nodes (plan steps), unlike traditional MCTS which typically uses simulations, **we employ a value model to evaluate the node’s value for efficiency**.
>    - For terminal nodes (detailed implementation), we parse the answer in the detailed implementation, compare it to the ground truth, and assign a value based on correctness
>
>
> - **Backup**: We backpropagate the value of newly expanded nodes to update the search tree.
>
> **Optimizing the Policy and Value Models.** After each round of MCTS, we optimize the policy model using SFT and the Step-APO algorithm. Additionally, we optimize the value model using Mean Squared Error (MSE) loss to ensure its outputs are closer to the state value $V$ of each node in the MCTS search tree. The updated policy and value models then generate higher-quality MCTS data.
>
> To further clarify the implementation of our plan-based MCTS, we have added ``Appendix F``, which provides a flowchart（``Figure 4``) and a detailed description of the process. Please kindly refer to the updated Appendix F for more information.

---

> ### Author Response · Authors · 2024-11-20
> **Response to Reviewer aihq (2/3)**
>
> > Q3: The comparison with baselines is insufficient, as it lacks search-based algorithms such as [1][2] in the baseline comparisons.
>
> Thank you for your valuable suggestion. CPL and the methods in [1, 2] share similarities in using MCTS to collect high-quality data for training. However, there are key differences:
>   - **Task-specific vs. High-Level Plans**: [1,2] search in the task-specific solution space, making it hard to generalize across tasks. In contrast, CPL searches on high-level plans in natural language, providing a universal interface across tasks. Applying RL to these general plans improves general reasoning and transfers gains across domains.
>   - **Supervision Method**: [1,2] use SFT to train policy, whereas CPL further apply Step-APO to leverage fine-grained supervision, boosting high-advantage plan steps (i.e., critical plan steps).
> We have included a discussion of these works in ``Appendix G``.
>
> Since the experimental setups of [1] and [2] differ from ours, it's difficult to rerun their methods with our model and data in the short term. However, to address your concern as best as we can, we used the **publicly available checkpoint**(alphazero-GSM8K) from [1] and evaluated it on out-of-domain (OOD) tasks, comparing it to [1]'s base model.  ([2] has not public any data/checkpoints). The results showed **a decrease in performance for some tasks (GPQA/BBH)**, proving that task-specific fine-tuning is difficult to generalize and can sometimes even hurt performance on other tasks. In contrast, CPL can improve performance across a wide range of OOD tasks.
>
> | Model                | Out-of-domain |           |         |         |           |
> |----------------------|---------------|-----------|---------|---------|-----------|
> |                      | HumanEval    | ARC-C     | GPQA    | BBH     | MMLU-stem |
> | Llama2              | 9.76         | 53.41     | 27.27   | 39.69   | 37.08     |
> | alphazero-GSM8K     | 10.37        | 54.27     | 23.74   | 25.77   | 37.49     |
>
>
> > Q4: Could you elaborate on the differences between training-based and testing-based approaches, as mentioned in [1][2]? If minimal resources are required at test time, is RL fine-tuning necessary?
>
> Thank you for your insightful question. We are pleased to elaborate on the relationship between training-based and test-time approaches.
>
> **Training-based (RL on self-generated data) vs. Test-time Scaling**:
> - Training-based methods require computational resources only periodically, while test-time scaling incurs high costs for each request, potentially impacting user experience.
> - As highlighted in [3], task-specific training can hinder performance on other tasks. Previous RL methods are often task-specific, making them struggle with OOD tasks. For example, in Alphazero[1] and AlphaMath[5] , training on math data decreased performance on other tasks. CPL addresses this by focusing RL on abstract plans rather than task-specific solutions, improving OOD generalization and enhances overall reasoning capabilities across tasks.
> - While test-time scaling improves performance, [4] points out its limitations: it is ineffective for simple and very difficult problems, and it also has a limit, with no further improvements once the scaling reaches a certain budget.
> - Both training-based and test-based methods are effective for improving reasoning and are not mutually exclusive. RL training significantly enhances test-time scaling by enabling the model to learn skills such as reflection, critique, and correction. Exploring how to apply RL to fully leverage the potential of test-time scaling, as well as optimizing test-time search efficiency, are crucial research topics.

---

> ### Author Response · Authors · 2024-11-20
> **Response to Reviewer aihq (3/3)**
>
> > Q5: Please provide more results for different sampling methods in Section 3.3. The authors only compared with CoT methods - how does it compare with MCTS without planning, Tree of Thoughts, Q*, and other similar methods?
>
> Thank you for your valuable suggestion. We are pleased to provide additional clarification.
>
> In Section 3.3, our primary goal is to compare **plan-based** versus **solution-based** search and training methods. We verified the advantages of plan-based methods **in repeated sampling**.
>
> Similar to repeated sampling, ToT, Q*, and MCTS **are all search algorithms**. In Section 3.3, our focus is on comparing the search content (i.e., high-level plans and task-specific solutions), which is not inherently tied to any specific search algorithm.
>
> To address your concern, we have supplemented our experiments with additional results comparing MCTS with planning and MCTS without planning on out-of-domain (OOD) tasks. The experimental results demonstrate that MCTS-based searching for high-level abstract plans consistently outperforms searching for task-specific solutions, showing stronger generalization capabilities.
>
> | Model                | Out-of-domain |           |        |
> |----------------------|---------------|-----------|--------|
> |                      | HumanEval    | ARC-C     | BBH    |
> | Base                | 40.90        | 52.05     | 58.79  |
> | w/o plan MCTS SFT   | 38.41        | 53.00     | 59.03  |
> | w/ plan MCTS SFT    | 42.68        | 54.44     | 59.68  |
>
> ---
> **Reference**
> [1] AlphaZero-Like Tree-Search can Guide Large Language Model Decoding and Training
> [2] Toward Self-Improvement of LLMs via Imagination, Searching, and Criticizing
> [3] Q*: Improving Multi-step Reasoning for LLMs with Deliberative Planning
> [4] Scaling LLM Test-Time Compute Optimally can be More Effective than Scaling Model Parameters
> [5] Chen, Guoxin, et al. "AlphaMath Almost Zero: process Supervision without process." NeurIPS 2024.
>
> &nbsp;
>
>
> We want to express our sincere gratitude for your review. If you have any further questions or concerns, please feel free to contact us at any time. We are always available and look forward to further discussions with you.
>
> Best regards,
> All Authors

---

> ### Author Response · Authors · 2024-11-24
> **Looking forward to your feedback**
>
> Dear Reviewer aihq,
>
> We hope this message finds you well. We sincerely appreciate your constructive feedback on our manuscript. Guided by your insightful suggestions, we have strived to address each point thoughtfully, which we believe have significantly improved the quality of our work.
>
> Thank you once again for your time and valuable guidance. We're eager to hear your thoughts on these revisions.
>
> Sincerely,
> All Authors

---

> ### Comment · Reviewer_aihq · 2024-11-28
>
> Thank you for the author's response, which addressed most of my concerns. I will maintain my score.

---

> > ### Author Response · Authors · 2024-11-29
> > **Thanks for your response**
> >
> > Dear Reviewer aihq,
> >
> > Thank you for your follow-up and for recognizing that we have addressed most of your concerns. Considering the efforts we made to resolve the raised issues, we sincerely hope you might reconsider and adjust the score. If there are any remaining questions or additional concerns, please do not hesitate to let us know—we are more than happy to address them.
> >
> > Sincerely,
> > All Authors

---

### Official Review · Reviewer_MKZR · 2024-11-03

**Soundness:** 1
**Presentation:** 3
**Contribution:** 1
**Rating:** 3
**Confidence:** 3

**Summary:**

The paper introduces Critical Plan Step Learning (CPL), a method designed to improve the generalization of large language models (LLMs) in multi-step reasoning tasks. The authors propose a novel approach that focuses on training LLMs through high-level abstract plans rather than task-specific actions. Their method consists of two main components: (1) using Monte Carlo Tree Search (MCTS) to explore diverse plan steps in multi-step tasks, and (2) introducing Step-level Advantage Preference Optimization (Step-APO), which refines the learning of critical plan steps by integrating advantage estimates from MCTS into Direct Preference Optimization (DPO). Experimental results show that the CPL method enhances both in-domain and out-of-domain reasoning capabilities across a variety of benchmarks, achieving significant improvements in datasets like GSM8K, MATH, HumanEval, and ARC-C.

**Strengths:**

- Approach: The paper proposes a method of focusing on high-level abstract plans instead of task-specific solutions, which helps improve model generalization across different reasoning tasks.
- Empirical Results: The experimental results showing significant improvements in both in-domain (GSM8K, MATH) and out-of-domain (HumanEval, ARC-C) reasoning benchmarks.
- Step-APO: The introduction of Step-APO, which integrates advantage estimates into DPO, provides a fine-grained optimization of critical plan steps.

**Weaknesses:**

The paper does not clearly define high-level abstract plans. How should one distinguish between abstract plans and task-specific plans? What are the differences between this method and the chain/tree/graph-of-thoughts approach? Additionally, the process of generating abstract plans is not described in detail, nor does the paper compare its method with existing works that use thoughts to enhance models' general reasoning abilities [1-2]. Furthermore, regarding Step-APO, several existing works have proposed very similar ideas [3-5], but the paper neither cites, discusses, nor compares its approach with these works. Overall, this work feels incomplete.

---

[1] Zelikman, Eric, et al. "Star: Bootstrapping reasoning with reasoning." NeurIPS 2022.

[2] Zelikman, Eric, et al. "Quiet-star: Language models can teach themselves to think before speaking." arXiv preprint arXiv:2403.09629 (2024).

[3] Rafailov, Rafael, et al. "From $ r $ to $ Q^* $: Your Language Model is Secretly a Q-Function." arXiv preprint arXiv:2404.12358 (2024).

[4] Zhong, Han, et al. "Dpo meets ppo: Reinforced token optimization for rlhf." arXiv preprint arXiv:2404.18922 (2024).

[5] Zeng, Yongcheng, et al. "Token-level Direct Preference Optimization." ICML 2024.

**Questions:**

See the Weaknesses section.

---

> ### Author Response · Authors · 2024-11-20
> **Response to Reviewer MKZR (1/2)**
>
> Thank you very much for your valuable feedback. We address your feedback point by point below.
>
> > Q1: The paper does not clearly define high-level abstract plans. How should one distinguish between abstract plans and task-specific plans?
>
> We are pleased to provide clarification. The definitions and comparison of abstract plans and task-specific solutions are explained in ``Lines 75-79`` and in ``Figure 2 (Left)``. You can also find our prompt and two-shot examples in ``Appendix A``, which are used to guide the model in generating the plan and solution.
>
> Abstract plans are **high-level strategies for solving problems in natural language, consisting of plan steps like determining applicable knowledge, breaking down problems, or analyzing key information.** In contrast, task-specific solutions, such as math computations or code, detail implementation steps derived from the plan. For example, outlining how to implement code is an abstract plan, while the code itself is a task-specific solution.
>
> Search and RL on abstract plans enhance generalization and diversity by focusing on high-level strategies rather than task-specific solutions, which often limit both exploration and generalization.
>
> > Q2: What are the differences between this method and the chain/tree/graph-of-thoughts approach?
>
> CPL shares similarities with these approaches in enhancing reasoning by generating thought processes, but they differ in the following ways:
> - CPL uses RL on self-generated plan thoughts to improve model general reasoning capabilities. CoT/ToT/PoT are different prompt techniques used during inference to guide the reasoning process.
>
> -  CoT/ToT/GoT's thoughts are task-specific. For example, in the Game of 24, a thought might be "4+9=13 (left:10 13 13)", indicating the current calculation and remaining numbers. RL on these task-specific thoughts makes it hard to generalize across tasks. In contrast, CPL generates high-level plans in natural language before detailed task solutions, providing a universal interface across tasks. Applying RL on such general plan thoughts enables the model to generalize better, improving reasoning and transferring these gains across domains.
>
> > Q3: Additionally, the process of generating abstract plans is not described in detail.
>
> We design a carefully crafted two-shot prompt (listed in ``Appendix A``) to guide the model in answering questions in two parts: (1) step-by-step plans, and (2) detailed implementation. In the MCTS search tree, each non-terminal node stores a single plan step, while terminal nodes store the detailed implementation of all preceding plan steps.
>
>
> To further clarify the implementation of our plan-based MCTS, we have added ``Appendix F``, which provides a detailed description of the process. Please kindly refer to the updated ``Appendix F`` for more information.
>
> > Q4: nor does the paper compare its method with existing works that use thoughts to enhance models' general reasoning abilities
>
> We appreciate your suggestion and are pleased to provide a comparison between our method and the works you mentioned, specifically STaR[1] and Quiet-STaR[2], which also use thoughts to enhance models' general reasoning abilities.
>
> The similarity of CPL with STaR and Quiet-STaR is that all methods aim to improve the model's reasoning capabilities by post-training on self-generated thoughts.
>
> The differences with STaR:
>
> -  **Task-specific vs. Plan-based Data**: STaR generates task-specific solution data, limiting generalization, while CPL generates plan-based data that aids in generalization.
>
> - **Repeated Sampling vs. MCTS**: STaR uses repeated sampling for thought generation, while CPL uses MCTS. MCTS explores good reasoning paths more efficiently and balances exploration with exploitation better than repeated sampling.
>
> - **Response-level vs. Step-level Supervision**: STaR provides response-level supervision, lacking fine-grained step-level signals, while CPL automatically obtains process supervision through MCTS and uses Step-APO for further improvement.
> We supplement STaR's experiments in the table below, demonstrating that CPL performs better on both in-domain and out-of-domain tasks.
>
> |                   | In domain |       | Out-of-domain |       |       |       |           |
> | :---------------: | :-------: | :---: | :-----------: | :---: | :---: | :---: | :-------: |
> |                   |   MATH    | GSM8K |   HumanEval   | ARC-C | GPQA  |  BBH  | MMLU-stem |
> | DeepseekMath-Base |   35.18   | 63.23 |     40.90     | 52.05 | 25.75 | 58.79 |   52.74   |
> |       STaR        |   37.68   | 70.13 |     43.29     | 52.73 | 27.78 | 60.45 |   54.20   |
> |        CPL        |   41.64   | 73.77 |     53.05     | 56.06 | 34.34 | 60.54 |   54.93   |

---

> ### Author Response · Authors · 2024-11-20
> **Response to Reviewer MKZR (2/2)**
>
> The differences with Quiet-STaR:
> While Quiet-STaR generates thoughts for each token during inference, we were unable to directly evaluate this method on our benchmarks. However, we highlight the following key differences:
> - **Efficiency**: During inference, Quite-STaR generates thoughts before each token of the solution, which results in substantial computational overhead. In contrast, CPL generates a complete, high-level plan of thoughts prior to producing the final solution, making CPL far more efficient.
> - **Guidance in Thought Generation**: CPL uses prompts to guide the LLM to generate abstract plans and reasoning related thoughts, while Quite-STaR generates the thoughts solely based on the preceding context, making CPL more capable of producing reasoning process.
> - **Supervision Approach**: Quite-STaR supervises thought generation based on whether they help with predicting the subsequent tokens, which leads to no further improvement after a few update steps. In contrast, CPL uses golden answers for more accurate supervision, enabling sustained improvement.
>
> > Q5: Furthermore, regarding Step-APO, several existing works have proposed very similar ideas [3-5], but the paper neither cites, discusses, nor compares its approach with these works.
>
> Thank you for your suggestion. We have included a discussion of these works in ``Appendix G``.
> These three works all provide a token-wise MDP formulation for RLHF. Specifically:
> - [3] does not propose a new RL algorithm but shows how to derive DPO in a token-level MDP and demonstrates that DPO can be viewed as an inverse Q-learning algorithm. CPL is **inspired by [3], and we cite [3] in our derivation process** (``line 230``).
> - TDPO[5] improves DPO by extending it to the token level and incorporates forward KL divergence constraints for each token, improving alignment.
> - RTO[4] models RLHF problems as a token-wise MDP, introducing the RTO algorithm, which learns a token-level reward function from preference data and performs policy optimization based on this learned reward signal.
>
> **Comparison of CPL with [4, 5]**:
> - **Step-level vs. Token-level MDPs**: CPL is based on step-level MDPs, while [4] and [5] are based on token-level MDPs. Step-level MDPs are better suited for capturing complex reasoning steps, whereas token-level MDPs focus more on each word choice during the generation process.
> - **Comparison with TDPO**: We compare CPL with TDPO (since RTO has not been open-sourced) and show that TDPO outperforms instance-level DPO, while Step-APO outperforms TDPO. This demonstrates that Step-APO is more suitable for complex reasoning tasks.
>
>
> |              | In domain |  |    Out-of-domain       |       |       |       |           |
> | :----------: | :-------: | :-----------: | :-------: | :---: | :---: | :---: | :-------: |
> |              |   MATH    |     GSM8K     | HumanEval | ARC-C | GPQA  |  BBH  | MMLU-stem |
> |     SFT      |   36.30   |     63.79     |   42.68   | 54.44 | 28.78 | 59.68 |   54.58   |
> | Instance-DPO |   37.72   |     69.29     |   43.90   | 54.61 | 24.24 | 60.13 |   54.42   |
> |     TDPO     |   39.12   |     69.90     |   48.78   | 54.61 | 28.29 | 59.94 |   54.08   |
> |   Step-APO   |   40.56   |     71.06     |   48.78   | 55.55 | 31.31 | 60.18 |   55.15   |
>
> ----
> **Reference**
> [1] Zelikman, Eric, et al. "Star: Bootstrapping reasoning with reasoning." NeurIPS 2022.
> [2] Zelikman, Eric, et al. "Quiet-star: Language models can teach themselves to think before speaking." arXiv preprint arXiv:2403.09629 (2024)
> [3] Rafailov, Rafael, et al. "From r to Q∗: Your Language Model is Secretly a Q-Function." arXiv preprint arXiv:2404.12358 (2024).
> [4] Zhong, Han, et al. "Dpo meets ppo: Reinforced token optimization for rlhf." arXiv preprint arXiv:2404.18922 (2024).
> [5] Zeng, Yongcheng, et al. "Token-level Direct Preference Optimization." ICML 2024.
>
> &nbsp;
>
> We want to express our sincere gratitude for your review. If you have any further questions or concerns, please feel free to contact us at any time. We are always available and look forward to further discussions with you.
>
> Best regards,
> All Authors

---

> > ### Author Response · Authors · 2024-11-24
> > **Looking forward to your feedback**
> >
> > Dear Reviewer MKZR,
> >
> > We hope this message finds you well. We sincerely appreciate your constructive feedback on our manuscript. Guided by your insightful suggestions, we have strived to address each point thoughtfully, which we believe have significantly improved the quality of our work.
> >
> > Thank you once again for your time and valuable guidance. We're eager to hear your thoughts on these revisions.
> >
> > Sincerely,
> > All Authors

---

> > > ### Comment · Reviewer_MKZR · 2024-11-26
> > >
> > > Thank you for your thoughtful response and the effort you have put into revising the manuscript. I appreciate the updates and the relevance of the prompts provided in the revised version. That said, I have decided to keep my score. My main concern remains whether the proposed plan introduces a fundamental distinction from existing approaches to thoughts. While the revisions address some aspects of this, I believe a more thorough analysis and clearer demonstration of the novelty would further strengthen the work. Thank you again for your efforts.

---

> > > > ### Author Response · Authors · 2024-11-26
> > > > **Response to Reviewer MKZR (2/2)**
> > > >
> > > > **Reference**
> > > >
> > > > [1] Wang, Chaojie, et al. "Q*: Improving multi-step reasoning for llms with deliberative planning." *arXiv* *preprint* *arXiv:2406.14283* (2024).
> > > > [2] Feng, Xidong, et al. "Alphazero-like tree-search can guide large language model decoding and training." ICML 2024.
> > > > [3] Chen, Guoxin, et al. "AlphaMath Almost Zero: process Supervision without process." NeurIPS 2024.
> > > > [4] Zelikman, Eric, et al. "Star: Bootstrapping reasoning with reasoning." NeurIPS 2022.
> > > > [5] Hwang, Hyeonbin, et al. "Self-Explore: Enhancing Mathematical Reasoning in Language Models with Fine-grained Rewards." EMNLP 2024.
> > > >
> > > > We sincerely hope that the above response addresses your concerns. If you have any further questions or concerns, please feel free to contact us at any time. We are always available and look forward to further discussions with you.
> > > >
> > > > Sincerely,
> > > > All Authors

---

> > > > ### Author Response · Authors · 2024-11-29
> > > > **Kindly request for your feedback**
> > > >
> > > > Dear Reviewer MKZR,
> > > >
> > > > Thank you for your follow-up and for recognizing that we have addressed some of your concerns. To further address your main concern, we have added a more thorough analysis and a clearer demonstration of the fundamental distinction between CPL and existing approaches. We believe this further highlights the novelty of CPL.
> > > >
> > > > We're eager to receive your feedback and sincerely hope you might reconsider and adjust the score. If you have any further concerns, please do not hesitate to let us know.
> > > >
> > > > Thank you again for your time and valuable feedback.
> > > >
> > > > Sincerely,
> > > > All Authors

---

> ### Author Response · Authors · 2024-11-26
> **Response to Reviewer MKZR (1/2)**
>
> Thank you for your time in providing us with feedback. We believe there may have been some misunderstanding regarding the core contributions and novelty of our work, and we are pleased to provide further clarification.
>
> > My main concern remains whether the proposed plan introduces a fundamental distinction from existing approaches to thoughts. While the revisions address some aspects of this, I believe a more thorough analysis and clearer demonstration of the novelty would further strengthen the work.
>
> The fundamental distinction between CPL and existing approaches lies in their focus: existing approaches prompt LLMs to directly generate task-specific solutions and perform **search-and-RL on these solutions**, whereas CPL prompts LLMs to first generate step-by-step plans along with corresponding solutions, performing **search-and-RL on natural language abstract plans as a universal interface across tasks**.
>
> In response to your previous Q1, we have already provided detailed explanations of plans and solutions, along with examples in `Figure 2` and the 2-shot examples in `Appendix A`. To further clarify, we present two additional examples below for your reference.
>
> - In the code task, "Iterate through the list, updating the largest number found so far" is a plan step, and the code `for num in numbers: \n if num > greatest: \n greatest = num` is a solution step.
> - In the math task, "Apply an LCM-GCD formula LCM(a,b)×GCD(a,b)=a×b to calculate the product of the two numbers" is a plan step, and "The product of the two numbers: a×b=36×6=216" is the corresponding solution step.
>
> **This design offers two key advantages**. First, it enables the model to develop more task-agnostic capabilities and generalize better. Second, CPL changes the search space from solution steps to abstract plan steps. Since different solutions may share the same underlying abstract plan, the model can concentrate on a smaller set of high-level plans, significantly reducing the search space.
>
> &nbsp;
>
> Next, we provide a thorough analysis and clearer demonstration of the novelty.
>
> ---
>
> **Research Significance:**
>
> There is a growing focus on using RL post-training to improve LLM reasoning capabilities. However, **a critical challenge is that obtaining accurate RL supervision can be difficult.** For example, in code, correctness evaluation is not as straightforward as in mathematics (where golden answers are available), and supervision based on test cases is scarce and costly. In contrast, domains like mathematics provide more accessible and accurate supervision. Given these challenges, **scaling RL across domains to build a general reasoner** becomes increasingly important. As the first work to address this issue, CPL offers a promising step in this direction.
>
> ---
>
> **Flaws in Existing Work:**
>
> Existing approaches focus on task-specific training, which significantly limits OOD generalization and the development of general reasoning capabilities. As highlighted in [1], **task-specific training can hinder performance on other tasks**.
>
> To reinforce this, we provided supplementary evidence below, which shows that **task-specific training on math tasks leads to performance drops in OOD tasks**.
>
> - AlphaZero[2]'s performance on GPQA decreased from 27.27 to 23.74, and BBH dropped from 39.69 to 25.77 (from Response to Reviewer aihq (2/3)).
> - AlphaMath[3]'s performance on BBH decreased from 58.79 to 56.63.(from `Table 1` in our paper)
> - Our newly included ablation study reveals that: MCTS on solutions dropped HumanEval performance from 40.90 to 38.41, whereas MCTS on plans improved it from 40.90 to 42.68 (from Response to Reviewer aihq (3/3)).
>
> ---
>
> **Exceptional OOD Results and Significance of CPL:**
>
> CPL addresses the above flaws by proposing using search-and-RL on abstract plans—in natural language as a universal interface across tasks—to enable the model to generalize effectively, rather than relying on task-specific solutions. Comprehensive experiments prove that **CPL consistently improves performance on OOD tasks, significantly outperforming task-specific training baselines** like STaR[4], AlphaMath[3], Self-Explore[5], and MCTS on solutions (from Response to Reviewer aihq (3/3)).
>
> |                   | HumanEval | ARC-C     | GPQA      |
> | ----------------- | --------- | --------- | --------- |
> | DeepseekMath-Base | 40.90     | 52.05     | 25.75     |
> | Self-Explore-MATH | 41.46     | 54.01     | 33.83     |
> | AlphaMath         | 49.39     | 53.41     | 33.33     |
> | STaR              | 43.29     | 52.73     | 27.78     |
> | CPL               | **53.05** | **56.06** | **34.34** |

---

### Official Review · Reviewer_db9Y · 2024-11-03

**Soundness:** 3
**Presentation:** 3
**Contribution:** 2
**Rating:** 6
**Confidence:** 2

**Summary:**

The paper proposes critical plan step learning method. The CPL approach does MCTS on the abstract level plan and develop diverse plans. And then they incorporate step level advantage preference optimization to learn policy effectively. Similar to DPO, they derive a step level APO objective. They demonstrate that the proposed algorithm improve the base model performance considerably on the in domain as well as out of domain tasks.

**Strengths:**

- Proposes a method for learning high level actions which author argue is good for generalization beyond in domain tasks
- The proposed algorithm is interesting and demonstrate improvement upon the base model performance.
- Their ablation on section 3.4 is insightful which shows improvement due to step-APO.

**Weaknesses:**

- LLM’s are quite stochastic how ever the paper only reports a single number in plots and table which does not provide a clear comparison across methods. I would appreciate if they could compare distribution (e.g., variance or so as well)
- Authors talk about scaling RL; but did not see efforts to address the sampling problem. I agree that search is reduced from MCTS over token level to abstract plan level. But the Isn’t the search space is still huge (theoretically of the same complexity as before)?
- The comparison is mainly around Deepseek model. I would suggest to try out another model for strengthen experimental conclusions

**Questions:**

- Rephrase: steps through Step-level Advantage Preference Optimization (Step-APO), which integrates advantage estimates for step preference obtained via MCTS into Direct Preference Optimization (DPO)
- Can you comment on your training times as well?
- I am curious how many samples are used for computing \pi_\ref (via sft?) and do you update it again?
- The difference with DPO is still not fully clear to me. In eqn 12 is the main difference that you do not need full trajectories but partial/split trajectories are also sufficient? But I can also do this with DPO, I believe.

---

> ### Author Response · Authors · 2024-11-20
> **Response to Reviewer db9Y (1/2)**
>
> Thank you very much for your valuable feedback. We address your feedback point by point below.
>
> > Q1: LLM’s are quite stochastic how ever the paper only reports a single number in plots and table which does not provide a clear comparison across methods. I would appreciate if they could compare distribution (e.g., variance or so as well)
>
> We use mainstream evaluation frameworks and follow the reporting methods of previous works [1,2].
> - For most experiments, we employed **greedy sampling** to avoid the randomness introduced by LLM sampling.
> - For the ARC-C, BBH, and MMLU-stem benchmarks, we used lm-evaluation-harness [3] for assessment. It employed the bootstrap method to calculate the standard deviation. This method involves repeatedly resampling the evaluation data with replacement to generate multiple "bootstrap samples" and computing metrics (e.g., accuracy) on each sample. From these results, we derived the standard deviation. The results and their associated standard deviations, presented below, validate the robustness of CPL’s performance across these benchmarks.
>
> | Model                   | ARC-C       | BBH          | MMLU-stem    |
> |-------------------------|-------------|--------------|--------------|
> | DeepseekMath-Base       | 52.05(1.46) | 58.79(0.56)  | 52.74(0.87)  |
> | Self-Explore-MATH       | 54.01(1.46) | 60.04(0.56)  | 54.04(0.88)  |
> | AlphaMath               | 53.41(1.46) | 56.63(0.57)  | 55.31(0.87)  |
> | CPL                     | 56.06(1.45) | 60.54(0.56)  | 55.44(0.87)  |
>
> > Q2: Authors talk about scaling RL; but did not see efforts to address the sampling problem. I agree that search is reduced from MCTS over token level to abstract plan level. But the Isn’t the search space is still huge (theoretically of the same complexity as before)?
>
> Let's first explain token-level and step-level search.
> - When using MCTS to explore LLM responses, token-level means that each token is treated as a node, and the potential token combinations result in an exponentially large search space, posing a significant challenge to the efficiency of MCTS. Replacing a single token with a step at each node can reduce the search space, as the depth of the step in a trajectory is much smaller than the number of tokens.
> - MCTS leverages the LLMs' generation probability to guide the selection process (i.e. PUCT), reducing the selection of unreasonable nodes. Handling multiple tokens in a node and using semantic constraints significantly reduces the search space.
>
> To further reduce the search space, we use prompts to guide the LLMs in generating high-level abstract plan steps. This changes the search space from solution steps to abstract plan steps. As different solutions may share the same underlying abstract plan, the model focuses on a smaller set of high-level plans, significantly reducing the search space.
>
> > Q3: The comparison is mainly around Deepseek model. I would suggest to try out another model for strengthen experimental conclusions.
>
> Thank you very much for your thoughtful suggestion.
>
> In addition to the DeepSeekMath, We ran CPL Round1 on Llama3 8B. The results are listed in the table below, demonstrating that CPL achieves significant gains for both OOD and in-domain tasks on Llama3. These findings confirm that CPL can effectively enhance the performance of models with varying capabilities, proving its robustness and general applicability across different models.
>
> | Model                   | In-domain      |               | Out-of-domain     |               |               |               |          |
> |-------------------------|----------------|---------------|-------|----------|--------|------|---------|
> |                         | MATH          | GSM8K         | HumanEval  | ARC-C         | GPQA          | BBH           | MMLU-stem     |
> | Llama-3-8B              | 18.16         | 49.17         | 37.80              | 57.94         | 27.27         | 62.92         | 55.82         |
> | Llama-3-8B + CPL (Round1)| 20.24         | 53.90          | 40.24             | 60.24         | 34.34         | 64.08         | 56.83         |

---

> ### Author Response · Authors · 2024-11-20
> **Response to Reviewer db9Y (2/2)**
>
> > Q4: Rephrase: steps through Step-level Advantage Preference Optimization (Step-APO), which integrates advantage estimates for step preference obtained via MCTS into Direct Preference Optimization (DPO)
>
> We have revised the sentence and have highlighted the change in blue in the updated manuscript. What we intended to convey is that Step-APO incorporates advantage estimates for step preference into the DPO, with these estimates being derived from MCTS.
>
> > Q5: Can you comment on your training times as well?
>
> We are pleased to provide details regarding the training times. All our experiments were conducted on 8 * H100 GPUs.
> - For MCTS, the data generation time for the first round, which involved 5k question-answer pairs and 200 iterations, was approximately 15 to 20 hours. In the second round, using 15k question-answer pairs and 100 iterations, the data generation time increased to around 2 days.
> - The SFT took around 30 minutes to 1 hour. The Step-APO took approximately 1 to 2 hours.
>
> > Q6: I am curious how many samples are used for computing $\pi_{ref}$ (via sft?) and do you update it again?
>
> For computing $\pi_{ref}$, we use the data generated in each round of MCTS to perform SFT on the base model. As a result, **the data used for each round of SFT changes accordingly**. In the first round, the number of training samples was 14k, while in the second round, it increased to 45k.
>
> > Q7: The difference with DPO is still not fully clear to me. In eqn 12 is the main difference that you do not need full trajectories but partial/split trajectories are also sufficient? But I can also do this with DPO, I believe.
>
> Thank you for your question. We are pleased to provide further clarification.
>
> DPO uses full trajectories and constructs pairs based on answer correctness, which lacks fine-grained step-level supervision. **The answer is yes that DPO can also work on partial trajectories, which is the step-DPO in our ablation study**. We list two losses for clear comparison.
>
> The Step-DPO loss:
> $\mathcal{L} _ \text{Step-DPO}(\pi _ {\theta}; \pi _ {ref}) = -\mathbb{E}{(s_t, a_t^w, a_t^l)\sim \mathcal{D}}\left[\log \sigma \left(\beta \log \frac{\pi _ {\theta}(a_t^w \mid s_t)}{\pi _ {ref}(a_t^w \mid s_t)} - \beta \log \frac{\pi_{\theta}(a_t^l \mid s_t)}{\pi _ {ref}(a_t^l \mid s_t)} \right) \right]$
>
> The Step-APO loss:
> $\mathcal{L} _ \text{Step-APO}(\pi _ {\theta}; \pi _ {ref}) = -\mathbb{E}{(s_t, a_t^w, a_t^l)\sim \mathcal{D}}\left[\log \sigma \left(\beta \log \frac{\pi _ {\theta}(a_t^w \mid s_t)}{\pi _ {ref}(a_t^w \mid s_t)} - V(s_{t+1}^w) - \beta \log \frac{\pi_{\theta}(a_t^l \mid s_t)}{\pi _ {ref}(a_t^l \mid s_t)} + V(s_{t+1}^l)  \right) \right]$
>
> Compare these two losses, the difference is that **step-DPO treats all pairs equally**. For example, a pair of win case (value=0.9) and loss case (value=-0.9) is given the same optimization weight as a win case (value=0.1) and loss case (value=-0.1). In contrast, **step-APO incorporates advantage differences calculated from the MCTS process**. This allows us to assign higher optimization weights to pairs with larger advantage differences, thus boosting high advantage plan steps (i.e., critical plan steps) and de-emphasizing erroneous ones.
>
> ---
> **Reference**
> [1] Chen, Guoxin, et al. "AlphaMath Almost Zero: process Supervision without process." NeurIPS 2024.
> [2] Hwang, Hyeonbin, et al. "Self-Explore: Enhancing Mathematical Reasoning in Language Models with Fine-grained Rewards." EMNLP 2024.
> [3] Gao, Leo, et al. "A framework for few-shot language model evaluation."
>
> &nbsp;
>
> We want to express our sincere gratitude for your review. If you have any further questions or concerns, please feel free to contact us at any time. We are always available and look forward to further discussions with you.
>
> Best regards,
> All Authors

---

> ### Author Response · Authors · 2024-11-24
> **Looking forward to your feedback**
>
> Dear Reviewer db9Y,
>
> We hope this message finds you well. We sincerely appreciate your constructive feedback on our manuscript. Guided by your insightful suggestions, we have strived to address each point thoughtfully, which we believe have significantly improved the quality of our work.
>
> Thank you once again for your time and valuable guidance. We're eager to hear your thoughts on these revisions.
>
> Sincerely,
> All Authors

---

> ### Comment · Reviewer_db9Y · 2024-11-30
>
> Thanks for addressing my comments and adding more details. I have no further questions.

---

### Author Response · Authors · 2024-11-20

Dear ACs and Reviewers,

Thank you very much for your thoughtful and constructive feedback. We would like to highlight the core contributions and significance of our work, which may have been overlooked.

Pretrained models are highly general, yet most existing RL-based reasoning improvement methods are task-specific. Our work offers valuable insights into training **general reasoners**. Specifically, we propose using search and RL on **plan thoughts—in natural language as a universal interface across tasks**—to enable the model to generalize effectively, rather than relying on task-specific solutions. Our experiments show that CPL training on math improves OOD task (e.g., code, science, commonsense).

CPL is a promising step toward scaling RL across domains. It also addresses the critical challenge of RL supervision, where **obtaining accurate signals can be difficult**. For instance, verifying code correctness can be costly, but tasks like math provide more accessible supervision. Using search-and-RL techniques with natural language plans on such tasks can generalize reasoning improvements to OOD tasks.

We addressed the main issues raised by reviewers and updated the pdf:
- **Verify CPL on another model (Llama3 8B)**: We conducted additional experiments with Llama3 8B, showing improvements on both OOD and in-domain tasks, which highlights the robustness and general applicability of CPL across models (added in `Appendix E`).
- **More baselines**:
  - Compared with STaR [1], showing CPL outperformed it on all tasks (added in `Table 1`).
  - Compared with TDPO [2], showing Step-APO outperformed it on all tasks (added in `Table 3`).
-  **More details about Plan-based MCTS**: We explained high-level abstract plans and how the plan is integrated into the MCTS (added in `Appendix F`).
-  **Supplementary discussion on related work** (added in `Appendix G`).
-  **Ablation study on plan-base learning**: Compared plan-based MCTS with non-plan-based MCTS, showing that plan-based MCTS consistently outperforms the non-plan-based approach on all tasks, proving the advantage of plan-based learning.

---
Reference

[1] Zelikman, Eric, et al. "Star: Bootstrapping reasoning with reasoning." NeurIPS 2022.
[2] Zeng, Yongcheng, et al. "Token-level Direct Preference Optimization." ICML 2024.

&nbsp;

Best regards,
All Authors

---

### Note · Authors · 2025-01-23

I have read and agree with the venue's withdrawal policy on behalf of myself and my co-authors.